# Fine control of metal concentrations is necessary for cells to discern zinc from cobalt

Deenah Osman [1,2], Andrew W. Foster[1,2], Junjun Chen[3], Kotryna Svedaite[1,2], Jonathan W. Steed [2], Elena Lurie-Luke[4], Thomas G. Huggins[3] & Nigel J. Robinson[1,2]

Bacteria possess transcription factors whose DNA-binding activity is altered upon binding to specific metals, but metal binding is not specific in vitro. Here we show that tight regulation of buffered intracellular metal concentrations is a prerequisite for metal specificity of Zur, ZntR, RcnR and FrmR in *Salmonella* Typhimurium. In cells, at non-inhibitory elevated concentrations, Zur and ZntR, only respond to Zn(II), RcnR to cobalt and FrmR to formaldehyde. However, in vitro all these sensors bind non-cognate metals, which alters DNA binding. We model the responses of these sensors to intracellular-buffered concentrations of Co(II) and Zn(II) based upon determined abundances, metal affinities and DNA affinities of each apo- and metalated sensor. The cognate sensors are modelled to respond at the lowest concentrations of their cognate metal, explaining specificity. However, other sensors are modelled to respond at concentrations only slightly higher, and cobalt or Zn(II) shock triggers mal-responses that match these predictions. Thus, perfect metal specificity is fine-tuned to a narrow range of buffered intracellular metal concentrations.

[1] Department of Biosciences, Durham University, Durham DH1 3LE, UK. [2] Department of Chemistry, Durham University, Durham DH1 3LE, UK. [3] Procter and Gamble, Mason Business Center, Cincinnati, OH 45040, USA. [4] Procter and Gamble, Singapore Innovation Center, Singapore 138589, Singapore. Correspondence and requests for materials should be addressed to N.J.R. (email: nigel.robinson@durham.ac.uk)

Most bacteria contain a set of metal sensors, each responding to a specific metal ion to modulate expression of genes encoding proteins involved in metal homoeostasis, which include transporters that either import specific metals during metal deficiency or export specific metals that are in excess. Correct regulation of metal homoeostasis is critical for a cell to achieve metal sufficiency while avoiding metal toxicity. Metal sensors are typically allosteric transcription factors whose DNA-binding activity is altered upon metal binding[1], resulting in metal-dependent modification of gene expression either via co-repression (as for Zur, Fig. 1a)[2, 3], co-activation for example via a conformational change which recruits RNA polymerase (as for ZntR, Fig. 1b)[4, 5] or de-repression (as for RcnR, Fig. 1c)[6]. A challenge exists because, in common with other proteins, metal sensors bind metal ions with an order of preference that matches the Irving–Williams series and are therefore not inherently selective for binding solely to their cognate metal (Supplementary Fig. 1)[7–10].

Protein mis-metalation is a feature of metal toxicity[11–14]. For example, Zn(II), cobalt (and copper) toxicity in *E. coli* involve mis-metalation of [4Fe-4S] clusters[15–17]. Metal sensors can also be mis-metalated in vivo, e.g., both the Mn(II)-sensor MntR and Fe(II)-sensor Fur from *Salmonella* Typhimurium strain 14028 can respond to both Mn(II) and Fe(II) in mutants lacking the cognate metal sensor[18]. In *Bacillus subtilis*, by contrast, Fe(II) is an antagonist to Mn(II) sensing by MntR, while Fur can again mal-respond to Mn(II) in mutants which overexpress Fur[19]. Exposure to Cd(II), Co(II) or Ni(II) dysregulates the expression of *czcD*, encoding a pneumococcal Zn(II)-efflux protein, suggesting mis-metalation of the Zn(II)-sensor SczA[20, 21]. Changes in transcription during short-term exposure to elevated metal concentrations, for example to a copper shock, have been distinguished from those occurring after longer-term steady-state adaptation to elevated copper; the latter characterised by selective expression of known copper resistance regulons[22]. Microbial susceptibility to metal fluxes is exploited by host immune systems to limit the growth of invading pathogens[23, 24]. Hosts restrict iron availability to pathogens[24], Zn(II) and manganese are withheld by calprotectin released from neutrophils, and there is also evidence of immune cell-mediated Zn(II) and copper toxicity[23–28]. Consistent with this, genes regulated by bacterial metal sensors are differentially expressed at specific sites and/or stages of infection, with an inability to mount an appropriate response rendering pathogens less virulent[24]. Metal sensors of pathogens may be adapted to transient metal fluxes, however, all bacterial metal sensors are predicted, and many have been observed, to bind non-cognate metals, at least in vitro[7, 10, 29–34]. It is hypothesised that the mis-sensing of metals may be a microbial 'Achilles heel', which has been exploited by immune systems.

The aim of this research was to understand how a bacterial cell selectively responds to Zn(II) and cobalt, and to discover whether metal sensing is liable to mal-respond to a wrong metal. The cognate sensors for Zn(II) (ZntR and Zur, Fig. 1a, b) and cobalt (RcnR, Fig. 1c), in the enteric pathogen *Salmonella* Typhimurium strain SL1344 (hereafter *Salmonella*) were previously identified[29]. Products of genes regulated by ZntR and Zur are, respectively, adapted to export and import Zn(II) and not cobalt, while RcnR-regulated RcnA is adapted to export cobalt and not Zn(II), in *Salmonella* and/or *E. coli*[29, 35–39]. By analogy to *E. coli*, *Salmonella* Zur is predicted to also regulate expression of an alternate ribosomal protein that does not require Zn(II), plus a periplasmic lysozyme inhibitor[2]. *Salmonella* also contains an RcnR-like sensor, FrmR (Fig. 1d), that is adapted to sense formaldehyde and does not respond during exposure to maximum non-inhibitory concentrations (MNICs) of metals, including cobalt and Zn(II)[29, 40]. However, FrmR unexpectedly binds and allosterically

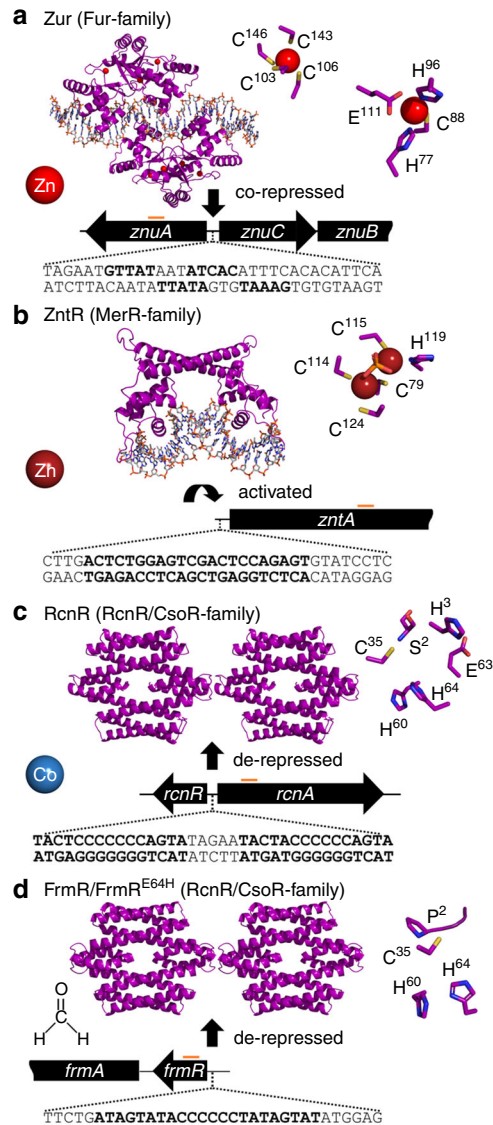

**Fig. 1** Zn(II), Co(II) and related formaldehyde sensors of *Salmonella*. Allosteric mechanisms of *Salmonella* sensors and structural models based on Protein Data Bank files 4MTD for Zur (**a**), 4WLW for ZntR (**b**), 5LCY for both RcnR (**c**) and FrmR$^{E64H}$ (**d**) with identified DNA-binding sites (bold), upstream of target genes. The DNA sequences shown were used for fluorescence anisotropy and orange bars indicate the region amplified by end point PCR and quantitative PCR. Known or inferred ligands for effector binding are enlarged: Zur contains a Cys$_4$-structural site and at least one sensory site. The dinuclear Zn(II) site of *E. coli* ZntR (PDB: 1Q08) is shown, noting that solution studies of *Salmonella* ZntR indicate a mononuclear site[5, 29]. An RcnR Co(II) site has been proposed, which may also include Glu$^{32}$[70]. FrmR$^{E64H}$ and FrmR have overlapping sites for formaldehyde (Cys$^{35}$, Pro$^{2}$) and metal binding (Cys$^{35}$, His$^{60}$ and either His$^{64}$ for FrmR$^{E64H}$ or Glu$^{64}$ for FrmR)[40]. Cognate effectors are depicted

responds to metals in vitro[29, 40], and a single amino acid substitution generates an FrmR variant (FrmR$^{E64H}$), which can respond to Zn(II) and cobalt in vivo[29]. Increased sensitivity to Zn(II) of FrmR$^{E64H}$ is due to an ~tenfold tighter Zn(II) affinity and ~4-fold weaker DNA affinity of apo-FrmR$^{E64H}$, relative to FrmR[29]. Thus, modest changes can generate a metal sensor from a non-metal sensor and this suggests that the cell may be poised close to thresholds for detecting and discerning between metals.

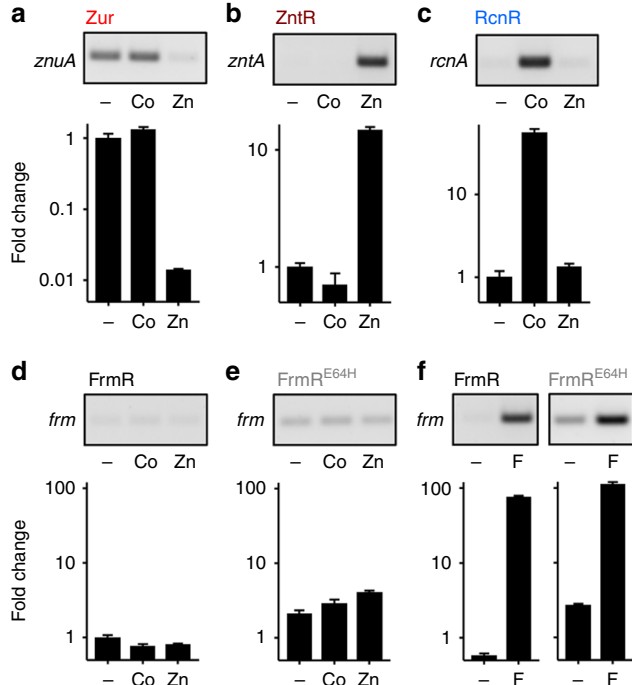

**Fig. 2** Each sensor responded specifically to one effector. Representative ($n = 3$) transcript abundance of *znuA* (regulated by Zur) (**a**), *zntA* (regulated by ZntR) (**b**), *rcnA* (regulated by RcnR) (**c**) and *frm* (regulated by FrmR or FrmR$^{E64H}$) (**d–f**), following growth of *Salmonella* in minimal media without supplementation (−) or with 0.25 μM cobalt (Co), 50 μM Zn(II) (Zn) or 50 μM formaldehyde (F; ~10% growth inhibition observed with each supplementation). Both end point PCR (upper) and qPCR (lower, error bars are s.d.) are shown. Quantitative data for regulation by FrmR and FrmR$^{E64H}$ is relative to the control without supplementation (−) in **d**. Data for control genes are presented in Supplementary Fig. 3 and full gel images in Supplementary Figs. 4 and 5

When the transcription of genes encoding Ni(II) import and export was engineered to rely on sensors adjusted to respond at higher Ni(II) concentrations, the cellular Ni(II) content increased relatively little and instead the sensors ceased to respond[41]. Thus, the sensitivity of a DNA-binding metal sensor is tuned to a buffered concentration of its cognate ion, presumably to regulate mechanisms that prevent this buffer from becoming depleted or saturated with metal[41]. Factors known to influence metal detection by each sensor are metal affinity, DNA affinity, the allosteric mechanism connecting metal binding to DNA binding, plus the abundance of sensor protein[1, 7, 10, 11, 19, 29, 30, 41–43]. Several of these parameters have been measured for different members of a set of sensors for Zn(II), Ni(II) and Co(II) in a common cell (*Synechocystis* PCC 6803)[7, 30, 31, 41]. The Ni(II) sensor, InrS, has the tightest affinity for Ni(II) while the Zn(II) sensor ZiaR has the greatest free energy coupling Zn(II) binding to DNA binding among the cells' set of sensors[7, 30]. This illustrates how metal selectivity can be understood by comparing the relative properties of different metal sensors within a common cell. Here such observations are further rationalised by relating the sensitivities of metal sensors to buffered concentrations of the respective ions.

Ultimately, metal sensitivity of a sensor will be some function of all of the above factors operating together. However, quantitatively combining all of these factors presents a challenge. In this work, we measure these parameters and incorporate them into mathematical models in order to understand in vivo specificity of sensors to Zn(II) and Co(II). The computational methods are set out in a format to assist their use by others. Sensors are modelled

to respond at lower-buffered concentrations of their cognate metal, compared to sensors for other effectors, explaining and correctly predicting metal specificity. However, sensors for other effectors are modelled to be only marginally less sensitive to the non-cognate metal, and indeed metal shock triggers predictable responses to non-cognate metals. Thus, we discover that tight regulation of buffered intracellular metal concentrations is a prerequisite for perfect metal specificity, rendering sensors vulnerable to dysregulation, with implications for the microbicidal action of metal fluxes.

## Results

**Sensors are selective at non-inhibitory concentrations**. To compare the response of Zur, ZntR, RcnR, FrmR and FrmR$^{E64H}$ to Zn(II) and cobalt, *Salmonella* cells were cultured in minimal media (~4–5 h) supplemented with MNICs of each metal (giving ≤10% inhibition of growth), ensuring multiple cell division cycles in the presence of metal (Supplementary Fig. 2). Transcript abundance was visualised by end point reverse-transcriptase PCR and enumerated by quantitative PCR (qPCR; Fig. 2a–e). Expression of *znuA* was repressed upon Zn(II) supplementation, but not upon cobalt supplementation, reflecting a selective response of Zur to Zn(II) (Fig. 2a). Under the same conditions, *zntA* transcripts accumulated in response to Zn(II) but not cobalt, confirming that ZntR-mediated activation of *zntA* expression was also specific for Zn(II) (Fig. 2b). Conversely, *rcnA* transcripts accumulated in response to cobalt but not Zn(II), confirming that repression of *rcnA* by RcnR was alleviated by cobalt, but not Zn(II) (Fig. 2c). The abundance of *frm* transcripts was monitored in *Salmonella* strains containing either FrmR or variant FrmR$^{E64H}$[29]. As expected, *frm* transcripts did not accumulate in response to cobalt or Zn(II) when regulated by FrmR, but surprisingly they also failed to respond when regulated by FrmR$^{E64H}$ (Fig. 2d, e), despite previous observations that this variant conferred cobalt and Zn(II)-dependent β-galactosidase activity[29]. Repression by both FrmR and FrmR$^{E64H}$ was exclusively alleviated by formaldehyde (Fig. 2f). Under these conditions, Zur, ZntR, RcnR and FrmR were selective for their cognate effector.

**Multiple sensors respond to cobalt shock**. Gene expression under the control of FrmR$^{E64H}$ was previously investigated via assays of β-galactosidase activity after short-term exposure to elevated cobalt: a cobalt shock[29, 40]. In an attempt to reconcile apparent differences between these past data and Fig. 2e, a logarithmically growing culture was exposed to cobalt for 10 min and transcript abundance visualised by end point PCR and enumerated by quantitative PCR (Supplementary Figs. 6 and 7 and Fig. 3). A twofold change in transcript abundance was used as a threshold for sensor responsiveness as indicated by arrows on quantitative PCR graphs throughout. Higher cobalt concentrations could be used during such short term, compared to prolonged, metal exposures with only a modest effect on cell viability observed (~15% reduction at 5 μM CoCl₂, Supplementary Fig. 11a), while these higher cobalt concentrations are inhibitory during prolonged exposure (~30% growth reduction at 5 μM CoCl₂, Supplementary Fig. 11b). Under these conditions, expression regulated by either RcnR or FrmR$^{E64H}$ was de-repressed by cobalt, while repression mediated by FrmR was unaffected, consistent with FrmR$^{E64H}$ being a cobalt-sensing variant of FrmR. However, during this transient cobalt exposure, *znuA* expression was also partially repressed relative to the control, and *zntA* expression activated (Fig. 3 and Supplementary Fig. 6), identifying both Zur and ZntR as targets of cobalt mis-metalation. The affinity of FrmR$^{E64H}$ for Co(II) is 500-fold weaker than that of RcnR and for this reason its response to

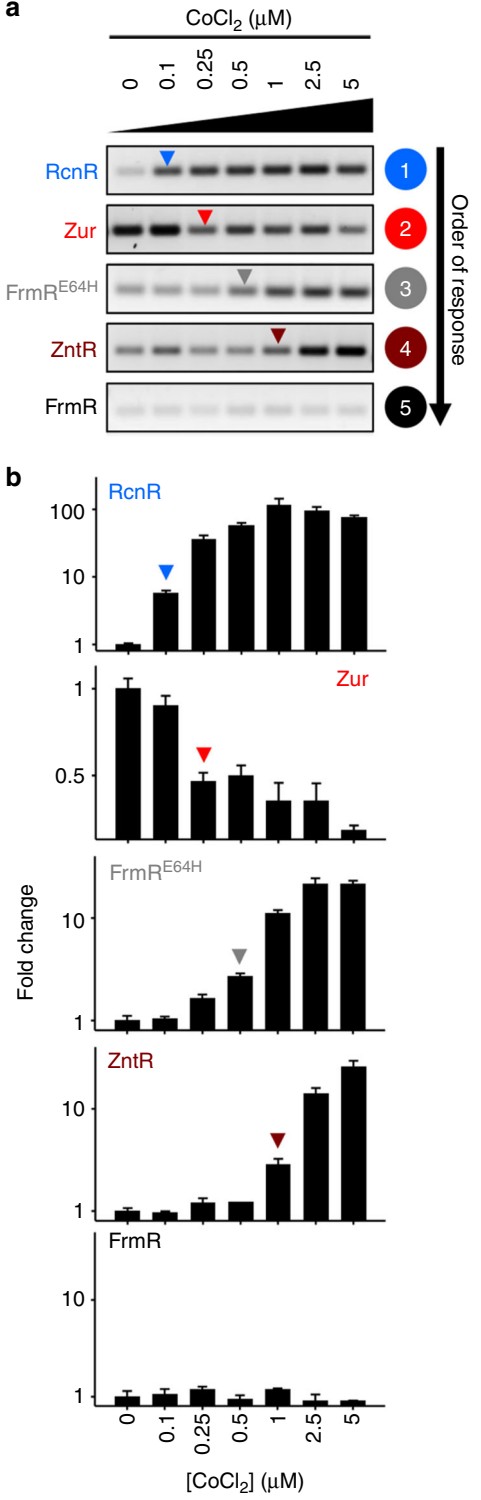

**Fig. 3** Cobalt shock triggers other sensors. **a** Representative ($n = 3$) transcript abundance following 10 min exposure of *Salmonella* to increasing [cobalt] assayed by end point PCR. Arrows identify the lowest exogenous [cobalt] at which each sensor appears to respond. Data for control genes are presented in Supplementary Fig. 8 and full gel images in Supplementary Fig. 9. **b** Transcript abundance for the samples shown in **a** measured by qPCR (error bars are s.d.). Arrows represent a ≥twofold change in transcript abundance. Heat maps of qPCR data from three biological replicates are presented in Supplementary Fig. 10

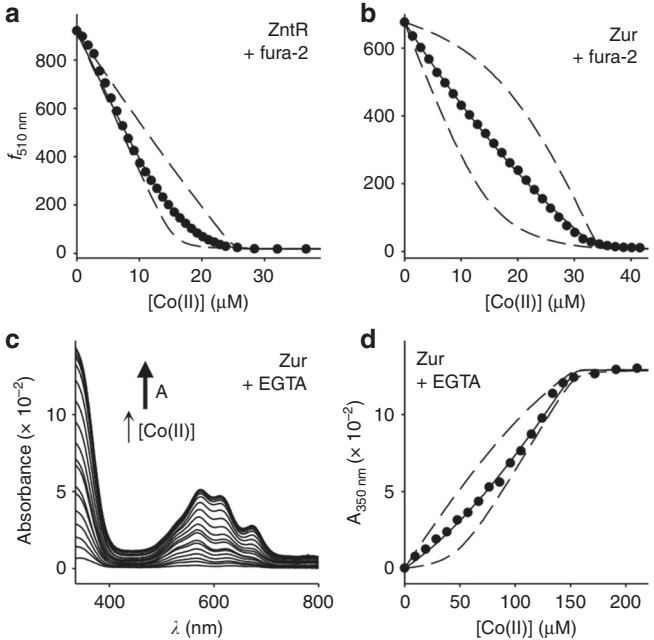

**Fig. 4** Co(II) affinities of ZntR and Zur. **a** Representative ($n = 3$) fura-2 fluorescence emission ($\lambda_{ex} = 360$ μM) upon titration of fura-2 (15.4 μM) and ZntR (9.8 μM) with Co(II). Solid line is a fit to a model describing competition from ZntR for one molar equivalent of Co(II). Dashed lines are simulated curves with $K_{Co(II)}$ tenfold tighter and tenfold weaker than the fitted value. **b** As in **a** but with fura-2 (14.6 μM) and Zur (9.8 μM) ($n = 5$). **c** Representative ($n = 3$) Zur absorbance spectra upon titration of Zur (52 μM) and EGTA (50 μM) with Co(II). **d** Binding isotherm at 350 nm for data shown in **c**. Solid line is a fit to a model describing competition from Zur for two molar equivalents of Co(II). Dashed lines are simulated curves with $K_{Co(II)}$ tenfold tighter and tenfold weaker than the fitted value

Co(II) was previously considered enigmatic[29]. However, whereas RcnR is tuned to a buffered concentration of Co(II) in cells grown in non-inhibitory cobalt concentrations, it is now evident that FrmR^E64H only responded during cobalt shock. Stepwise regulation of *Salmonella* sensors in response to increasing concentrations of cobalt shock occurred in the order RcnR, Zur, FrmR^E64H, ZntR, followed by FrmR, which did not respond (Fig. 3). It is proposed that non-cobalt sensors responded under cobalt shock because the cytosolic buffer became transiently saturated and higher intracellular concentrations occurred. This is consistent with growth inhibition during prolonged exposure to these higher metal concentrations (Supplementary Fig. 11b).

**ZntR and Zur bind Co(II) with sub-micromolar affinities**. We have previously demonstrated that in vitro both ZntR and Zur bind Co(II) in sites that can be displaced by Zn(II)[29]. To determine their Co(II) affinities (hereafter affinity refers to a dissociation constant), both proteins were purified following overexpression in *E. coli* and confirmed to be ≥95% pure (Supplementary Fig. 12) and ≥90% reduced[29]. ZntR was ≥95% metal free, and Zur contained ~1 molar eq. Zn(II) consistent with filling of a structural Zn(II) site identified in Zur and other Fur-family members[44]. Both proteins were competed against the fluorophore fura-2, which exhibits a decrease in fluorescence emission upon Co(II) binding and has been used to determine Co(II) affinities of metal sensor proteins including RcnR and FrmR^E64H (Fig. 4a, b)[29,31]. ZntR binds two Co(II) ions per dimer[29], and both sites were observed during competition with fura-2, with a combined affinity of $9.5 (\pm 1.0) \times 10^{-8}$ M (Fig. 4a and Table 1). Zur binds up to four Co(II) or Zn(II) ions per dimer in addition to the structural

**Table 1 Metal affinities, DNA affinities, allosteric coupling-free energies and abundance of *Salmonella* sensors**

| Sensor | Metal | Metal affinity (M) | DNA affinity (M) | $\Delta G_C$ (kcal mol$^{-1}$) | Abundance (assemblies cell$^{-1}$)[a] |
|---|---|---|---|---|---|
| Zur | — | n.a. | $\geq 2.7\ (\pm 0.4) \times 10^{-5}$ | n.a. | 21 ($\pm 7$) |
| | Co(II) | $1.5\ (\pm 0.6) \times 10^{-8}$ | $3.1\ (\pm 0.3) \times 10^{-8}$ | $\leq -4.0\ (\pm 0.1)$ | |
| | Zn(II)$_4$ | n.a. | $5.4\ (\pm 1.8) \times 10^{-8}$ | $\leq -3.7\ (\pm 0.2)$ | |
| | Zn(II)$_2$ | $6.4\ (\pm 0.4) \times 10^{-13\text{b}}$ | $4.1\ (\pm 1.0) \times 10^{-8}$ | $\leq -3.9\ (\pm 0.2)$ | |
| ZntR | — | n.a. | $1.1\ (\pm 0.3) \times 10^{-6}$ | n.a. | 34 ($\pm 15$) |
| | Co(II) | $9.5\ (\pm 1.0) \times 10^{-8}$ | $3.4\ (\pm 1.0) \times 10^{-7}$ | $-0.7\ (\pm 0.2)$ | |
| | Zn(II) | $3.2\ (\pm 0.7) \times 10^{-12\text{b}}$ | $6.5\ (\pm 3.3) \times 10^{-7}$ | $-0.3\ (\pm 0.2)$ | |
| RcnR | — | n.a. | $1.5\ (\pm 0.8) \times 10^{-7\text{c}}$ | n.a. | 22 ($\pm 2$) |
| | Co(II) | $5.1\ (\pm 0.9) \times 10^{-10\text{b}}$ | $\geq 1.5\ (\pm 0.2) \times 10^{-5\text{c}}$ | $\geq +2.7\ (\pm 0.2)^{\text{c}}$ | |
| | Zn(II) | $9.4\ (\pm 1.0) \times 10^{-12}$ | $\geq 1.3\ (\pm 0.2) \times 10^{-5}$ | $\geq +2.6\ (\pm 0.1)$ | |
| FrmR$^{\text{E64H}}$ | — | n.a. | $4.3\ (\pm 0.4) \times 10^{-7\text{b}}$ | n.a. | 149 ($\pm 4$)$^{\text{b}}$ |
| | Co(II) | $2.6\ (\pm 0.4) \times 10^{-7\text{b}}$ | $2.3\ (\pm 0.3) \times 10^{-6}$ | $+1.0\ (\pm 0.1)$ | |
| | Zn(II) | $2.3\ (\pm 0.3) \times 10^{-11\text{b}}$ | $3.5\ (\pm 0.7) \times 10^{-6\text{b}}$ | $+1.2\ (\pm 0.2)^{\text{b}}$ | |
| FrmR | — | n.a. | $9.9\ (\pm 0.3) \times 10^{-8\text{b}}$ | n.a. | 135 ($\pm 17$)$^{\text{b}}$ |
| | Co(II) | $7.6\ (\pm 0.4) \times 10^{-6\text{b}}$ | n.d.$^{\text{d}}$ | n.d. | |
| | Zn(II) | $1.7\ (\pm 0.7) \times 10^{-10\text{b}}$ | $3.1\ (\pm 0.4) \times 10^{-6\text{b}}$ | $+2.0\ (\pm 0.1)^{\text{b}}$ | |

All constants and abundances are means of at least triplicate determinations ('n' specified in figure legends) with ± s.d.
*n.a.* not applicable, *n.d.* not determined
[a]Number of functional unit per cell. For Zur and ZntR, this is a dimer and for RcnR, FrmR and FrmR$^{\text{E64H}}$, this is a tetramer
[b]These values were determined previously[29]
[c]These values were determined previously[40]
[d]A value of $2.0 \times 10^{-6}$ M was used for mathematical modelling, estimated as described in the text

Zn(II) sites (a total of 6:1 Me(II):Zur dimer)[29]. Competition with fura-2 did not distinguish between four Co(II)-binding events (Fig. 4b). In contrast, Zn(II) binding by Zur occurs with strong negative cooperativity: Sites 1–2 are tighter than site 3 by ~120-fold and site 3 is tighter than site 4 by ~6000-fold[29]. The intense UV-visible absorption spectra of Co(II) binding to Zur (replicated here in Supplementary Fig. 13a)[29], allowed observable competition with the spectrally silent chelator ethylene glycol tetraacetic acid (EGTA) (Fig. 4c and Supplementary Fig. 13b), and confirmed the lack of detectable cooperativity of Co(II) binding to Zur. Data were fit to a model describing four combined Co(II)-binding events per Zur dimer (Fig. 4d). The affinity of Zur for Co(II) was determined to be $1.5\ (\pm 0.6) \times 10^{-8}$ M averaged from both fura-2 and EGTA competition experiments (Table 1). The affinity of ZntR for Co(II) was comparable to that of FrmR$^{\text{E64H}}$, while the affinity of Zur was approximately one order of magnitude tighter than either ZntR or FrmR$^{\text{E64H}}$.

**Co(II) affects DNA binding by Zur, FrmR$^{\text{E64H}}$ and ZntR.** Since Zur, FrmR$^{\text{E64H}}$ and ZntR responded to cobalt shock in vivo (Fig. 3 and Supplementary Figs. 6 and 10), it seemed probable that Co(II) triggers allosteric responses which promote association of Zur with the *znuA* operator–promoter, dissociation of FrmR$^{\text{E64H}}$ from the *frmRA* operator–promoter, and activation by ZntR of the *zntA* operator–promoter. The degree to which metal binding is coupled to DNA binding can be described as the allosteric coupling-free energy ($\Delta G_C$), which, in combination with metal affinity, contributes to metal selectivity[1, 7, 42]. Fluorescence anisotropy, using a fluorescently labelled dsDNA fragment containing the identified Zur-binding site upstream of *znuA* (Fig. 1a), was used to examine the effect of Co(II) on allostery. Initially, the stoichiometry of Zur binding to DNA was determined with saturating concentrations of Zn(II) (ensuring filling of exchangeable sites 1–4) and demonstrating that two Zn(II)-Zur dimers bind to the *znuA* operator–promoter sequence (Fig. 5a). *E. coli* Zur (93% identity to *Salmonella* Zur) binds to a similar target DNA sequence as two adjacent dimers with positive cooperativity[2]. The data for *Salmonella* Zur were fit to a model describing sequential binding of two Zur dimers to the *znuA* operator–promoter. The DNA affinity of both Zn(II)-Zur dimers was determined to be $5.4\ (\pm 1.8) \times 10^{-8}$ M (Fig. 5b and Table 1).

In contrast, apo-Zur (with only the structural Zn(II) sites filled) bound to the *znuA* operator–promoter with an affinity weaker than Zn(II)-Zur by ~500-fold ($\geq 2.7\ (\pm 0.4) \times 10^{-5}$ M) (Fig. 5b and Table 1). The free energy coupling metal binding to DNA binding, $\Delta G_C$, for Zn(II)-Zur was calculated to be $\leq -3.7\ (\pm 0.2)$ kcal mol$^{-1}$. Importantly, Co(II) also promoted DNA binding by Zur with DNA affinity and $\Delta G_C$ values of Co(II)-Zur determined to be $3.1\ (\pm 0.3) \times 10^{-8}$ M and $\leq -4.0\ (\pm 0.1)$ kcal mol$^{-1}$, respectively (Fig. 5c and Table 1). Thus, Co(II) was as effective as Zn(II) in activating the allosteric mechanism of Zur.

FrmR$^{\text{E64H}}$ was purified and biochemically characterised as described for ZntR and Zur (Supplementary Fig. 12). Fluorescence anisotropy using a fluorescently labelled dsDNA fragment containing the *frmRA* operator–promoter (Fig. 1d), confirmed that Co(II) triggered an allosteric response by FrmR$^{\text{E64H}}$ (Fig. 5d and Table 1), such that the DNA affinity of Co(II)-FrmR$^{\text{E64H}}$ was ~fivefold weaker than apo-FrmR$^{\text{E64H}}$ with $\Delta G_C$ + 1.0 ($\pm 0.1$) kcal mol$^{-1}$ (Table 1). Apo-ZntR bound a fluorescently labelled dsDNA fragment containing the *zntA* operator–promoter (Figs. 1b and 5e). At least two DNA-binding events were observed and the anisotropy change associated with the first (tighter) binding event was determined (Fig. 5e). A similar change in anisotropy ($\Delta r_{\text{obs}}$ ~0.025) was independently modelled for binding of an apo-ZntR dimer to the *zntA* operator–promoter sequence, and a DNA affinity of $1.1\ (\pm 0.3) \times 10^{-6}$ M was determined (Fig. 5f and Table 1). An equivalent complex of Zn(II)-ZntR bound the target DNA with an affinity of $6.5\ (\pm 3.3) \times 10^{-7}$ M (Fig. 5g and Table 1), revealing a free energy coupling metal binding to DNA binding, $\Delta G_C$, for Zn(II)-ZntR of $-0.3\ (\pm 0.2)$ kcal mol$^{-1}$. In *E. coli*, ternary DNA complexes containing two ZntR dimers occur at high ZntR concentrations[45], and additional increases in $\Delta r_{\text{obs}}$ at elevated ZntR concentrations suggest that *Salmonella* ZntR forms similar ternary complexes (Fig. 5e, g). Co(II) promoted similar changes in DNA binding with affinity and $\Delta G_C$ values determined to be $3.4\ (\pm 1.0) \times 10^{-7}$ M and $-0.7\ (\pm 0.2)$ kcal mol$^{-1}$, respectively (Fig. 5h and Table 1). Both Co(II) and Zn(II) encouraged formation of ternary complexes, at least on this (34 bp) DNA fragment (Fig. 5g, h).

**Similar numbers of ZntR, RcnR and Zur protomers per cell.** The number of copies of FrmR and FrmR$^{\text{E64H}}$ per cell was

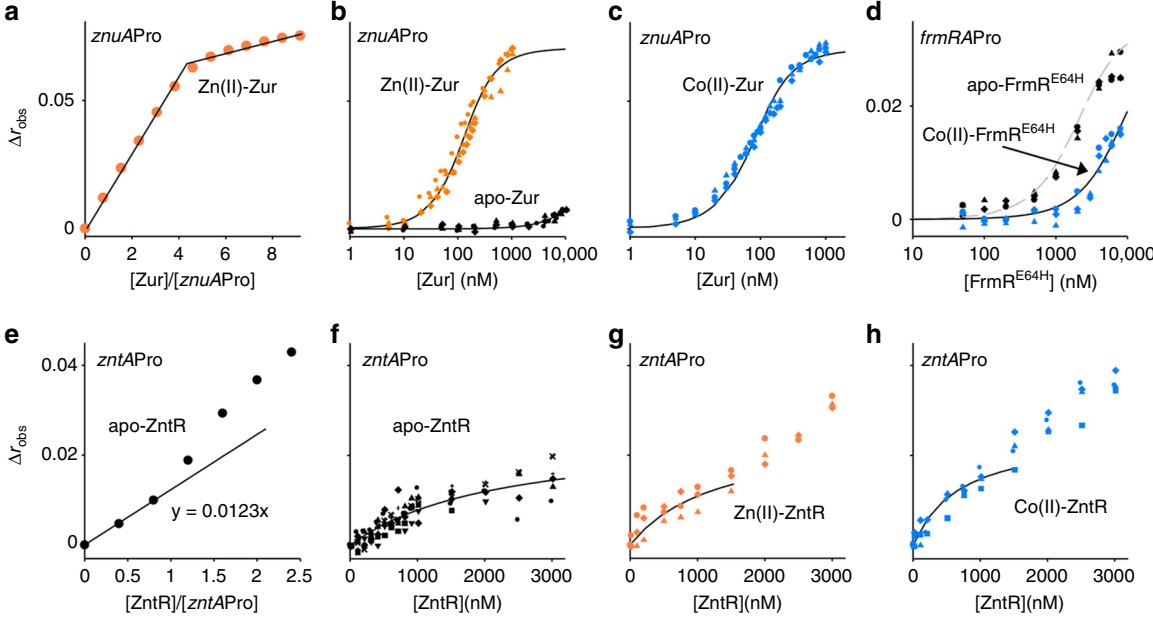

**Fig. 5** DNA affinities of Zur, Co(II)-FrmR[E64H] and ZntR. **a** Anisotropy change upon titration of 1 µM *znuA*Pro with Zn(II)-Zur in the presence of 1 µM Zn(II). **b** As **a** but with 10 nM *znuA*Pro and Zn(II)-Zur in the presence of 1 µM Zn(II) (orange symbols, $n = 3$), or apo-Zur with 5 mM EDTA (black symbols, $n = 3$). Symbol shapes represent individual experiments. Data were fit to a model describing a 2:1 Zur dimer (non-dissociable):DNA stoichiometry and lines are simulated curves using the mean DNA affinity across the experiments shown. **c** As **b** but with Co(II)-Zur and 1 µM Co(II) ($n = 3$). **d** Anisotropy change upon titration of *frmRA*Pro (10 nM) with apo-FrmR[E64H] in 5 mM EDTA (black symbols, $n = 3$) or Co(II)-FrmR[E64H] in 100 µM Co(II) (blue symbols, $n = 3$). Co(II)-FrmR[E64H] data were fit to a model describing a 2:1 FrmR[E64H] tetramer (non-dissociable):DNA stoichiometry and solid line is a simulated curve using the mean DNA affinity across the experiments shown[29, 40]. Dashed grey line is a simulated curve using the published DNA affinity for apo-FrmR[E64H] [7]. **e** As **a** but with apo-ZntR and 2.5 µM *zntA*Pro in 5 mM EDTA. Line is a linear fit to the first three data points predicting $\Delta r_{obs} = 0.0246$ at 2:1 ZntR:*zntA*Pro. **f** As **b** but with apo-ZntR and 10 nM *zntA*Pro ($n = 7$). **g** As **f** but with Zn(II)-ZntR ($n = 3$). **h** As **f** but with Co(II)-ZntR in 5 µM Co(II) ($n = 4$). Data in **f**, **g** and **h** were fit to a model describing 1:1 ZntR dimer (non-dissociable):DNA and lines are simulated curves using the mean DNA affinity across the experiments shown. For **g** and **h**, to determine the DNA affinity of the tightest binding event, data to 1500 nM ZntR monomer were used

determined previously[29]. Similar measurements were made for *Salmonella* Zur and ZntR using quantitative multiple reaction monitoring (MRM) tandem mass spectrometry (Table 1, Supplementary Fig. 14 and Supplementary Tables 1–3): These abundances were 21 (±7) Zur dimers per cell and 34 (±15) ZntR dimers per cell (Table 1). Quantification of RcnR was initially challenging due to interfering species from the complex *Salmonella* cell lysates. RcnR was therefore enriched in a quantitative manner via partial purification. The abundance of RcnR was thus determined to be 22 (±2) tetramers per cell (Table 1).

**Thermodynamic data predict the responses to cobalt.** Fractional metal saturation of a sensor (determined by $1/K_1$ alone, Fig. 6) has commonly been used as a surrogate measure of metal sensitivity[10]. However, metal binding and DNA binding are thermodynamically coupled such that DNA occupancy is not fully represented by $1/K_1$ alone (Fig. 6)[1, 46]. A complete data set that includes determined metal affinity, DNA affinity of apo-sensor, DNA affinity of metalated sensor ($1/K_1$, $1/K_3$ and $1/K_4$) and cellular abundance of RcnR, Zur, ZntR, FrmR[E64H] and FrmR (Table 1), should enable fractional occupancy of the respective operator–promoters to be calculated as a function of buffered Co(II) concentrations. To improve upon our recent attempts to combine such data sets[29, 41], we developed a script for use in Dynafit[47], to simultaneously fit the equilibria shown in Fig. 6 (Supplementary Software). To mathematically define the concept of a cellular-buffered metal concentration, a hypothetical buffer with a defined metal affinity ($1/K_5$) was introduced into the calculations. $[M]_{total}$ was maintained >1000-fold higher than $[P]_{total}$, and tenfold lower than $[B]_{total}$ to ensure a surplus of available metal at a defined (buffered) concentration. $1/K_5$ was altered,

iteratively, to achieve a [buffered metal] range from $10^{-3}$ to $10^{-16}$ M (Supplementary Data 1, Supplementary Software and Methods section).

RcnR was thus modelled to respond at the lowest cobalt concentration explaining why this is the bona fide sensor of Co(II) (Fig. 7). Because the weak Co(II) affinity of FrmR precluded determination of the DNA affinity for Co(II)-FrmR, this was estimated from $1/K_4$ for Zn(II)-FrmR and the fold-difference between $1/K_4$ for the Co(II)-FrmR[E64H] and Zn(II)-FrmR[E64H] variant (Table 1)[29]. Fractional DNA occupancy by FrmR[E64H] did not reach that of FrmR, due to the weaker DNA affinity of apo-FrmR[E64H] relative to apo-FrmR (Fig. 7)[29]. The metalated form of MerR-family regulators (Fig. 1b), such as ZntR, activate expression by distorting their target promoter[48, 49], therefore, *zntA*Pro bound by Co(II)-ZntR ((P•M)•D) was used to represent the active species. This implied a dynamic range close to that of FrmR[E64H] (Fig. 7). All of the other sensors were shown to be tuned above the cobalt sensitivity of RcnR, which would avoid mal-responses to Co(II). However, the margin for specificity was narrow such that Zur would also respond to Co(II) if the concentration became an order of magnitude greater than the set point for RcnR. To create the perfect metal selectivity observed in Fig. 2, there must be fine control of the intracellular cobalt concentration: Under cobalt shock, such control became imperfect (Fig. 3). Moreover, the observed sequence of activation of each sensor in response to increasing cobalt shock agreed with the order predicted by the thermodynamic models (Figs. 3 and 7): Noting that the modelled responses of FrmR[E64H] and ZntR overlapped (Fig. 7), although expression data indicated that the former was more sensitive to cobalt than the latter, perhaps because the effects of Co(II) on DNA binding do not fully reflect

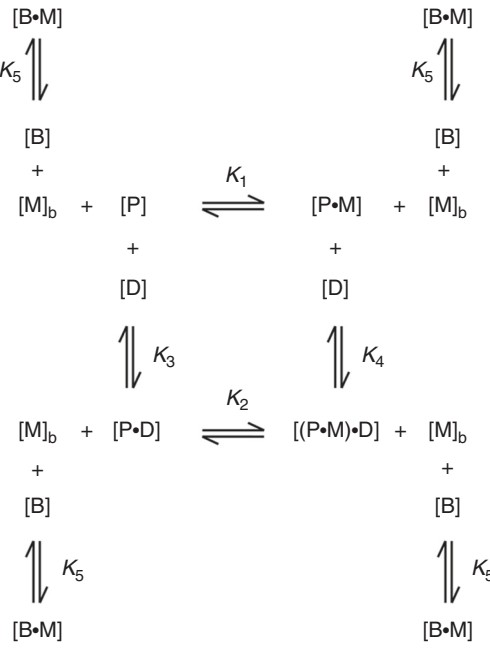

**Fig. 6** Thermodynamic coupling of metal and DNA binding. Four allosteric conformations (end-states) typical of metal sensor proteins: apo-protein (P), metal-protein (P•M), apo-protein-DNA (P•D) or metal-protein-DNA ((P•M)•D). Dynafit was used to simultaneously model these coupled equilibria to determine the fractional occupancy of each operator–promoter with sensor ((P•D + (P•M)•D)/$D_{total}$), as a function of buffered metal concentration ($M_b$) (see 'Methods' section, Supplementary Data 1 and Supplementary Software). $K_1$–$K_5$ are association constants. Buffered metal was achieved by including a hypothetical buffer component (B) with defined metal affinity ($1/K_5$)

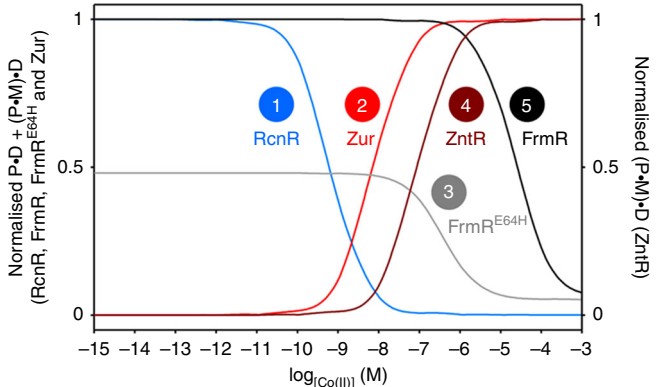

**Fig. 7** Calculated responses to intracellular Co(II). Calculated fractional occupancy ((P•D + (P•M)•D)/$D_{total}$) of DNA targets with RcnR (blue line), Zur (red line), FrmR[E64H] (grey line) or FrmR (black line), or of (P•M)•D/$D_{total}$ (dark red line) for Co(II)-ZntR, as a function of buffered [Co(II)], determined using Co(II) affinities, DNA affinities and abundance values in Table 1. FrmR and FrmR[E64H] were normalised to the same scale. De-repression by RcnR, FrmR and FrmR[E64H] occurs as the fractional occupancy of their promoters decrease, co-repression by Zur occurs as occupancy of its promoter increases, and activation by ZntR occurs as the fractional occupancy of its promoter with metalated ZntR increases. Numbering reflects the order of response observed for each sensor in Fig. 3

ZntR activation (Fig. 3). Supplementary Data 1 and Supplementary Software include a Dynafit script and template Excel spreadsheet to enable calculation of promoter occupancy by a metal sensor as a function of buffered metal concentration, when values of $K_1$, $K_3$, $K_4$, $[D]_{total}$ and $[P]_{total}$ (Fig. 6), are known or can be estimated (detailed step-by-step instructions are described in the 'Methods' section).

**Thermodynamic data predict the responses to Zn(II).** FrmR[E64H] shares at least four metal-binding ligands with RcnR[29] (Fig. 1c, d); therefore, we hypothesised that the Zn(II) affinity of RcnR may be comparable to FrmR[E64H]. RcnR was purified (Supplementary Fig. 12), and found to withhold one molar equivalent of Zn(II) from the spectral Zn(II)-probe magfura-2 consistent with four Zn(II)-binding sites per RcnR tetramer and an affinity at least tenfold tighter than that of magfura-2 (magfura-2 $K_{Zn(II)} = 2 \times 10^{-8}$ M) (Fig. 8a). Consequently, to accurately determine the RcnR Zn(II) affinity, a Zn(II) probe with a tighter Zn(II) affinity was used. Competition against quin-2 ($K_{Zn(II)} = 3.7 \times 10^{-12}$ M) revealed an affinity of RcnR for Zn(II) across four sites of $9.4 (\pm 1.0) \times 10^{-12}$ M (Fig. 8b and Table 1), which was ~2.5-fold tighter than FrmR[E64H]. The DNA affinity of Zn(II)-RcnR for the *rcnA* operator–promoter (Fig. 1c) was determined to be $\geq 1.3 (\pm 0.2) \times 10^{-5}$ M, which is $\geq$90-fold weaker than apo-RcnR revealing that Zn(II) induced an allosteric change that weakened DNA binding (Fig. 8c and Table 1). Notably, the $\Delta G_C$ of Zn(II)-RcnR was comparable to Co(II)-RcnR[40].

The DNA affinity of Zur is tightened when all four exchangeable Zn(II) sites are filled (Fig. 5b), however, the weakest sites have Zn(II) affinities in the region of $10^{-7}$ M. Using fluorescence anisotropy, we confirmed that filling of the tightest two sites alone (referred to as Zn(II)$_2$-Zur; $K_{Zn(II)} = 6.4 (\pm 0.4) \times 10^{-13}$ M, Table 1), was sufficient to induce an allosteric change that enabled Zur to bind to the *znuA* operator–promoter. The DNA affinity of Zn(II)$_2$-Zur was $4.1 (\pm 1.0) \times 10^{-8}$ M (Fig. 8d, and Table 1). Using the Zn(II) affinities, DNA affinities and abundance of Zur, ZntR, RcnR, FrmR[E64H] and FrmR (Table 1), the fractional occupancy of their respective operator–promoters (with total protein or with Zn(II)-ZntR) were modelled as a function of buffered Zn(II) concentration using the same procedures as described for Co(II) (Fig. 9a). Analogous to the models for Co(II) (Fig. 7), the cognate sensors for Zn(II) were calculated to respond at the lowest buffered Zn(II) concentrations, once again explaining specificity, in this case for Zn(II) (Fig. 9a). Sensors for other effectors were tuned above this concentration, however the margin for specificity was again narrow such that RcnR would also respond to Zn(II) if the concentration became an order of magnitude greater than the set point for ZntR. To create the perfect metal selectivity shown in Fig. 2, as with Co(II), there must also be fine control of intracellular Zn(II) concentrations.

To investigate the response of each sensor to Zn(II) shock, *Salmonella* cells were exposed to increasing Zn(II) concentrations for 10 min (Fig. 9b, c and Supplementary Fig. 11c). The highest Zn(II) concentrations (80 and 100 μM) were inhibitory during prolonged exposure (Supplementary Fig. 11d). Under these conditions, the stepwise pattern of Zn(II)-responsive gene expression, either monitored by end point PCR or qPCR, again aligned with the thermodynamic models, such that Zur responded at the lowest Zn(II) concentrations, followed by ZntR, RcnR and FrmR[E64H], and lastly FrmR which did not respond in vivo (Fig. 9a–c). Notably, the thermodynamic models were equivocal with respect to the relative sensitivities of RcnR and FrmR[E64H] to Zn(II) with the curves intersecting (Fig. 9a), and for these two sensors the order observed by quantitative PCR or end point PCR was indistinguishable (Fig. 9b, c), both were less sensitive than the bona fide Zn(II) sensors and more sensitive than FrmR. Selectivity is adapted to operate perfectly only when

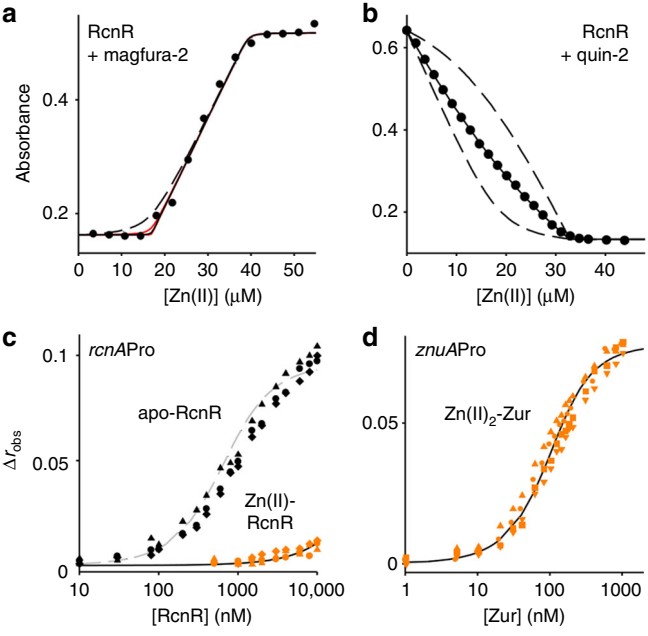

**Fig. 8** Zn(II) affinity of RcnR and effects of Zn(II) on DNA binding. **a** Representative ($n = 4$) magfura-2 absorbance at 325 nm upon titration of magfura-2 (22.6 μM) and RcnR (16.9 μM) with Zn(II). Solid black line is a fit to a model describing competition from RcnR for one molar equivalent of Zn(II). Solid red lines are simulated curves with $K_{Zn(II)}$ tenfold tighter (not visible) and tenfold weaker than the fitted value. Dashed line is a simulated curve with $K_{Zn(II)}$ 100-fold weaker. **b** Representative ($n = 4$) quin-2 absorbance at 265 nm upon titration of quin-2 (18.0 μM) and RcnR (14.9 μM) with Zn(II). Solid line is a fit to a model describing competition from RcnR for one molar equivalent of Zn(II). Dashed lines are simulated curves with $K_{Zn(II)}$ tenfold tighter and tenfold weaker than the fitted value. **c** Anisotropy change upon titration of *rcnA*Pro (10 nM) with apo-RcnR in the presence of 5 mM EDTA (black symbols, $n = 3$), or Zn(II)-RcnR (orange symbols, $n = 3$). Symbol shapes represent individual experiments. Data for Zn(II)-RcnR were fit to a model describing a 2:1 RcnR tetramer (non-dissociable):DNA stoichiometry and solid line is a simulated curve using the mean DNA affinity across the experiments shown[40]. Dashed grey line is a simulated curve describing the published apo-RcnR DNA affinity for comparison[40]. **d** Anisotropy change upon titration of *znuA*Pro (10 nM) with Zn(II)$_2$-Zur (orange symbols, $n = 5$). Symbol shapes represent individual experiments. Data were fit to a model describing a 2:1 Zur dimer (non-dissociable):DNA stoichiometry and lines are simulated curves using the mean DNA affinity across the experiments shown

cells exert fine control over intracellular metal concentrations and after Zn(II) shock, such control becomes imperfect. The modest differential between the fractional occupancy curves for the different sensors reveals that cells are on the cusp of mis-sensing Zn(II), as well as Co(II), when subjected to metal shocks (Figs. 7 and 9a).

## Discussion

Our calculations of gene regulation at different intracellular Co(II) and Zn(II) concentrations explain metal selectivity in metal sensing in terms of equilibrium thermodynamics by using determined metal affinities, DNA affinities, coupling free energies and the number of sensor molecules per cell (Figs. 7 and 9a). At first inspection, these models seem incorrect by revealing that RcnR is inherently more sensitive to Zn(II) than to Co(II) by one to two orders of magnitude (Supplementary Fig. 18), yet RcnR showed the opposite selectivity in cells during prolonged growth in elevated metal and is known to be a Co(II) sensor (Figs. 1

and 2)[29]. The explanation is that metal sensors are tuned to the buffered concentrations of their cognate metal[41], and the buffered concentration of Zn(II), but not cobalt, is maintained below the set point for RcnR. The set points for the Zn(II) sensors ZntR and Zur reveal this lower buffered concentration for Zn(II) (Fig. 9a). In this context, metal specificity now becomes readily understandable by comparing the sensitivities for Zn(II) (Fig. 9a), and for Co(II) (Fig. 7), of the five sensors to reveal that the bona fide sensors are the most sensitive in the set. During prolonged growth in elevated Zn(II), the intracellular Zn(II) concentration must have been finely controlled to within about one order of magnitude in order to trigger Zur and ZntR, but not RcnR or FrmR$^{E64H}$ (Figs. 2 and 9a). This must be a buffered Zn(II) concentration, with associative metal transfer, since one hydrated ion per cell volume equates to ~$10^{-9}$ M, which would be sufficient to trigger RcnR and FrmR$^{E64H}$ (Fig. 9a)[50]. Similarly, during prolonged growth in elevated Co(II), the intracellular buffered Co(II) concentration must also have been finely controlled to within about one order of magnitude in order to trigger RcnR but not Zur (Figs. 2 and 7). Thus, these metal sensors are adapted to discriminate perfectly between these inorganic elements only when metals are buffered, with associative metal transfer, and when metal concentrations are finely controlled.

The models predict that if the buffer becomes saturated then the Zn(II) sensors will respond to Co(II) and vice versa the Co(II) sensor will respond to Zn(II) (Figs. 7 and 9a). During Zn(II) shock, FrmR$^{E64H}$ and RcnR did respond, consistent with the Zn(II) concentration having transiently increased above the buffered concentration (Fig. 9b, c). Similarly during Co(II) shock Zur, FrmR$^{E64H}$ and ZntR responded (Fig. 3). For both metals, the order of the response to increasing metal shock (Fig. 3 and 9b, c), correlated with the order predicted from the thermodynamic properties of the sensors (Figs. 7 and 9a), further validating the models and suggesting that even the shock responses are not solely determined by on-rates (kinetics). This also reveals that when assigning metal specificity to metal sensors by monitoring gene expression in cells exposed to exogenous metals, care should be taken to optimise the growth conditions and avoid saturation of the intracellular buffer. For sensors with more than one DNA target, multiple set points may exist to allow a graded response to changing metal demands as the cytosolic metal buffer becomes increasingly full. Intriguingly, *Bacillus subtilis* Zur has at least three set-points reflecting filling of its multiple Zn(II) sites and its varying DNA affinities on different operator–promoters, giving rise to three waves of Zn(II)-dependent gene expression[51]. These three waves could reflect different levels of saturation of the cytosolic buffer. Alternatively, evidence of mal-responses of sensors for other metals might indicate if one of the waves occurs when the buffer becomes fully Zn(II) saturated.

The crowded cytosol contains a multitude of sulphur, nitrogen and oxygen ligands associated with an array of metabolites and macromolecules, many of which can be readily organised into different metal-binding combinations and geometries. Such a polydisperse mixture will inevitably bind and buffer metals in the order of the Irving Williams series[41]. It is anticipated that cytosol-facing metal-binding sites of metal transporters will also be tuned to these buffered metal concentrations. For some metals, and in some organisms, the cytosolic buffer may be dominated by a single molecule such as glutathione or its substitutes such as bacillithiol, or a free amino acid such as histidine[41, 52–55]. Macromolecules with multiple labile metal sites, e.g., metallothioneins, may also be induced to expand the depth of the buffer when metals such as Zn(II) increase in abundance[56]. At present, it is unclear whether or not the buffer for either Co(II) or Zn(II) in *Salmonella* is dominated by a single, and potentially shared, molecule. Notably, mutants deficient in the synthesis of

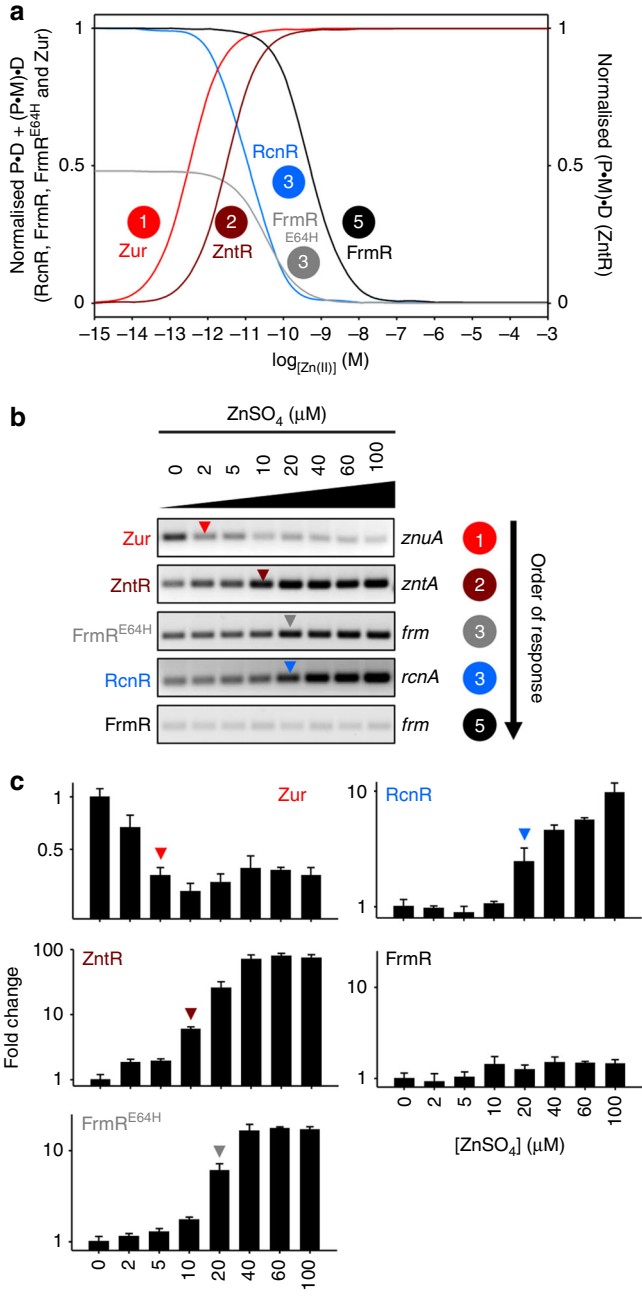

**Fig. 9** Predicted and observed responses to Zn(II). **a** Calculated fractional occupancy $((P \bullet D + (P \bullet M) \bullet D)/D_{total})$ of DNA targets with RcnR (blue line), Zur (red line), FrmR[E64H] (grey line) or FrmR (black line), or of $(P \bullet M) \bullet D/D_{total}$ (dark red line) for Zn(II)-ZntR, as a function of buffered [Zn(II)], determined using Zn(II) affinities, DNA affinities and abundance values in Table 1. Numbering reflects the order of response visualised for each sensor in Fig. 9b, c. **b** Representative ($n = 3$) transcript abundance following 10 min exposure of *Salmonella* to increasing [Zn(II)] assayed by end point PCR. Arrows identify the lowest observed exogenous [Zn(II)] at which each sensor appeared to respond. Data for control genes are presented in Supplementary Fig. 15 and full gel images in Supplementary Fig. 16. **c** Transcript abundance for the samples shown in **b** measured by qPCR (error bars are s.d.). Arrows represent a ≥twofold change in transcript abundance. Heat maps of qPCR data from three biological replicates are presented in Supplementary Fig. 17

glutathione showed impaired detection of Co(II) and Zn(II) by FrmR[E64H][29], but now we know that metal detection by this variant sensor only occurs during metal shock, presumably once components of the bona fide buffer have become saturated. In contrast, there is negligible effect of glutathione on Zn(II) sensing by *Salmonella* ZntR[29], whereas Zn(II) sensing by *B. subtilis* CzrA is enhanced in the absence of bacillithiol[52]. It is noteworthy that *Salmonella* is at least an order of magnitude more sensitive to exogenous cobalt than Zn(II) (Supplementary Fig. 11). Unlike *E. coli*, *Salmonella* requires cobalt to synthesise cobalamin, vitamin B$_{12}$, but this is only made under anaerobic conditions which perhaps renders *Salmonella* proteins susceptible to mis-metalation by unwanted, un-sequestered, cobalt during aerobic growth. In eukaryotes, there is scope for diversity in buffered metal concentrations within different intracellular compartments (nucleus, organelles, vesicles, trans-Golgi network, endoplasmic reticulum, for example) and spectral probes have been developed to interrogate such concentrations[57–59]. Since metal-sensing transcriptional regulators can also report upon metal occupancy in vivo and are tuned to metal concentrations for example in the bacterial cytosol, then their metal sensitivities provide an alternative approach to interrogate the vital buffered metal concentrations: In short, the $K_5$ values at which each sensor responds using the calculations described here (Fig. 6, Supplementary Data 1, Supplementary Software and Methods section). In the same way that metal selectivity of metal sensors becomes comprehensible in the context of these values (Figs. 7 and 9a and Supplementary Fig. 18), so metalation of other metalloproteins should become understandable once a complement of cellular $K_5$ values for different metals have been calculated.

In conclusion, we discovered that perfect metal specificity in metal sensing was restricted to a finely controlled range of buffered metal concentrations, which were exceeded during metal shocks (Figs. 3, 7 and 9). These data support the prediction that bacteria are susceptible to the mis-sensing of metals and hence the notion that this vulnerability is exploited by immune systems. Metals, chelants and ionophores also have a long history of use as antimicrobials in medicine, agriculture and as preservatives[60, 61]. However, the development of this wide spectrum of metal related treatments has largely been empirical, up to now.

## Methods

**Bacterial strains, DNA manipulations and growth conditions**. *S. enterica* sv. Typhimurium strain SL1344 was used as wild type and strain LB5010a was used as a restriction-deficient modification proficient host for DNA manipulations. Both were a gift from J.S. Cavet (University of Manchester), and originally from the *Salmonella* Genetic Stock Centre[29]. Deletion derivative Δ*frmR* (SL1344 lacking the *frmR* coding sequence) was generated previously[29]. *E. coli* strain DH5α was used for routine cloning and strain BL21(DE3) was used for recombinant protein overexpression (both from historical lab stocks). *E. coli* strains BW25113Δ*zntR::kan* (JW3254-5) and BW25113Δ*nikR::kan* (JW3446-3) were originally from the Keio collection[62] (and a gift from D. Weinkove, Durham University). Kanamycin resistance cassettes were removed using the helper plasmid pCP20 carrying the FLP recombinase. Promoter-*lacZ* fusion constructs with the *frmR* operator–promoter and *frmR* or *frmR[E64H]* coding sequence upstream of *lacZ*, have been described previously[29]. Bacteria were cultured aerobically (shaking at 200 rpm) at 37 °C in LB medium or M9 minimal medium, supplemented with thiamine (0.001%, w/v) and either L-histidine (20 µg ml$^{-1}$) for *Salmonella* or 1 µM ferric citrate for *E. coli*. Kanamycin (25 µg ml$^{-1}$), chloramphenicol (8 µg ml$^{-1}$) and carbenicillin (100 µg ml$^{-1}$) were added where appropriate. Maximum non-inhibitory concentrations (MNIC; defined as the maximum concentration which inhibited growth by ~10 %) of CoCl$_2$ and ZnSO$_4$ were determined in M9 minimal medium supplemented with metal salt, (following dilution of overnight cultures to an OD of 0.025 at 600 nm. For metal shock exposures, logarithmic cells were statically cooled to 25 °C for 20 min followed by a 10 min exposure to CoCl$_2$ or ZnSO$_4$ before dilution in phosphate-buffered saline and enumeration on LB agar. Concentrations of metal salts were confirmed by ICP-MS.

**Generation of *E. coli* BW25113 double-deletion mutants**. BW25113Δ*zur::cat* was generated by the λ Red method[63], using plasmid pKD3 and primers 1 and 2 (Supplementary Table 4; hereafter all primer numbers relate to this table). Mutants

were selected on LB medium supplemented with chloramphenicol. The Δzur::cat fragment was moved into strain BW25113ΔzntR (kan cassette removed) by P1 transduction. The chloramphenicol resistance cassette was removed and genotype confirmed by PCR using primers 3–6. P1 transducing lysate from BW25113ΔrcnR:: kan (JW2092-1, a gift from P. Chivers, Durham University) was used to move the ΔrcnR::kan fragment into BW25113ΔnikR (kan cassette removed). The kan cassette was removed and genotype confirmed by PCR using primers 7–10.

**RNA extraction and reverse-transcriptase PCR.** Expression mediated by FrmR and FrmR[E64H] was measured in Salmonella strain SL1344ΔfrmR containing either $P_{frmRA}$-frmR or $P_{frmRA}$-frmR[E64H] reporter constructs (generating SL1344[FrmR] and SL1344[FrmRE64H], respectively) cultured in supplemented M9 minimal medium following dilution of overnight cultures to an OD of 0.025 at 600 nm. To enable direct comparison of metal sensor responses with FrmR[E64H]-mediated regulation, expression of rcnA, znuA and zntA was measured in SL1344FrmR[E64H], with the exception of Supplementary Fig. 6, where SL1344[FrmR] was used as a further control. The medium was supplemented with MNICs of metals or formaldehyde and grown to mid-logarithmic phase prior to assays. MNICs under these growth conditions were 0.25 μM CoCl₂, 50 μM ZnSO₄ (described above; Supplementary Fig. 11), and 50 μM formaldehyde (determined under the same conditions[40]). For metal shock exposures, logarithmic cells were statically cooled to 25 °C for 20 min followed by a 10 min exposure to CoCl₂ or ZnSO₄. An aliquot (1.2 ml) of culture was used for RNA extraction using RNeasy® Protect Bacteria Mini Kit (Qiagen) as described by the manufacturer. RNA was quantified by absorbance at 260 nm, and treated with DNase I (Fermentas; 1 U per 44 ng RNA), and 300 ng RNA used per reverse transcriptase reaction (ImProm-II™ Reverse Transcription System, Promega). Negative controls without reverse transcriptase were performed in parallel.

**Transcript abundance by end point and quantitative PCR.** Transcript abundance was assessed using primers 11 and 12 for rcnA, 13 and 14 for znuA, 15 and 16 for zntA, 17 and 18 for lacZ, 19 and 20 for rrsD, and 21 and 22 for rpoD, each pair designed to amplify an ~150 bp fragment (Supplementary Table 4). For end point PCR, fragments were subsequently resolved by agarose gel electrophoresis. Gels were imaged with a Gel-Doc XR + gel documentation system (Bio-Rad). Quantitative PCR analyses were performed in 20 μl reactions using 2 ng of cDNA as a template, 0.8 μM of the appropriate primer pairs and PowerUp™ SYBR® Green Master Mix (ThermoFisher Scientific). Each sample was analysed in three technical replicates using a Rotor-Gene Q 2plex (Qiagen). The fold change in transcript level relative to control conditions was analysed using the $2^{-\Delta\Delta CT}$ method with rpoD as the reference gene[64]. Trends were confirmed with biological replicates on three occasions.

**Protein overexpression and purification.** E. coli BL21(DE3) containing pETzntR, pETzur, pETfrmR[E64H] and pETrcnR was used to overexpress ZntR, Zur, FrmR[E64H] and RcnR, respectively[29]. Protein purification was conducted using a combination of Ni(II) affinity, gel filtration, heparin affinity and ion-exchange chromatography[29]. Experimentally determined extinction coefficients were used to quantify purified proteins[29]. Proteins were confirmed to be ≥95% pure as assessed by SDS-PAGE (Supplementary Fig. 12). Anaerobic protein stocks (maintained in an anaerobic chamber) were prepared as described and confirmed to be ≥95% metal free and ≥90% reduced[29], with the exception of Zur which contained ~1 molar equivalent of Zn(II) (per monomer) as purified. All in vitro experiments were carried out under anaerobic conditions using Chelex-treated and N₂-purged buffers[29].

**Determination of metal affinities.** All experiments were conducted in 100 mM NaCl, 400 mM KCl, 10 mM HEPES pH 7.0. For competition with fura-2, CoCl₂ was titrated into a mixed solution of protein and fura-2 and fluorescence emission was recorded at equilibrium at 510 nm ($\lambda_{ex} = 360$ nm; $T = 20$ °C) using a Cary Eclipse fluorescence spectrophotometer (Agilent Technologies)[29, 31]. Fura-2 was quantified using the extinction coefficient $\varepsilon_{363\,nm} = 28,000$ M$^{-1}$ cm$^{-1}$ [31]. For competition with EGTA, CoCl₂ was titrated into a mixed solution of Zur and EGTA, and absorption spectra were recorded at equilibrium using a λ₃₅ UV-visible spectrophotometer (Perkin Elmer Life Sciences). Control experiments without EGTA were also performed. For competition with magfura-2 or quin-2, ZnCl₂ was titrated into a mixed solution of RcnR and magfura-2 or RcnR and quin-2, and absorbance was recorded at equilibrium at 325 nm (magfura-2) or 265 nm (quin-2). Magfura-2 and quin-2 were quantified using the extinction coefficients $\varepsilon_{369\,nm} = 22,000$ M$^{-1}$ cm$^{-1}$ [65] and $\varepsilon_{261\,nm} = 37,000$ M$^{-1}$ cm$^{-1}$ [66], respectively. Competition data were fit to models described in figure legends and Table 1 using Dynafit to determine Co(II) and Zn(II) affinities[47]. Mean and standard deviation values were determined from at least triplicate analyses ('n' specified in figure legends). Fura-2 $K_{Co(II)} = 8.64 \times 10^{-9}$ M at pH 7.0[67], EGTA $K_{Co(II)} = 7.89 \times 10^{-9}$ M at pH 7.0 determined using Schwarzenbach's α co-efficient method[68], magfura-2 $K_{Zn(II)} = 2 \times 10^{-8}$ M at pH 7.0[69] and quin-2 $K_{Zn(II)} = 3.7 \times 10^{-12}$ M[66].

**Fluorescence anisotropy.** Fluorescently labelled double-stranded DNA probes, znuAPro and zntAPro were generated using oligonucleotides 23 (hexachloro-fluorocein labelled) and 24 containing the identified Zur binding site upstream of znuA, or 25 (hexachlorofluorocein labelled) and 26 containing the identified ZntR-binding site upstream of zntA. frmRAPro and rcnAPro have been described previously[29, 40]. All oligonucleotides are listed in Supplementary Table 4. Complementary single-stranded oligonucleotides were annealed by heating a mixture containing 10 or 20 μM of each oligonucleotide to 95 °C in 30 mM NaCl, 120 mM KCl, 10 mM HEPES pH 7.0 and cooling in a thermal cycler at −0.5 °C per minute to 10 °C. Fluorescently labelled annealed probes were analysed by native PAGE (12% w/v). All experiments were conducted in 60 mM NaCl, 240 mM KCl, 10 mM HEPES pH 7.0, with inclusion of 100 μM CoCl₂ for Co(II)-FrmR[E64H], 1 μM ZnCl₂ for Zn(II)-Zur, 1 μM CoCl₂ for Co(II)-Zur and 5 μM CoCl₂ for Co(II)-ZntR. DNA binding by apo-RcnR, apo-FrmR[E64H], apo-Zur and apo-ZntR was performed with addition of 5 mM EDTA. Zn(II)-RcnR was prepared in 200 mM NaCl, 800 mM KCl, 10 mM HEPES pH 7.0 with addition of 1.2 molar equivalents of ZnCl₂ (per monomer). Zur, ZntR and FrmR[E64H] were prepared in 100 mM NaCl, 400 mM KCl, 10 mM HEPES pH 7.0, with addition of 2.2 molar equivalents ZnCl₂ (per monomer) for Zn(II)-Zur, 1 molar equivalent ZnCl₂ for Zn(II)₂-Zur, which only saturates two of the sites, 2.2 molar equivalents CoCl₂ for Co(II)-Zur, 1.2 molar equivalents ZnCl₂ or CoCl₂, for Zn(II)- and Co(II)-ZntR, respectively, or 100 μM CoCl₂ for Co(II)-FrmR[E64H]. Experiments to determine the stoichiometry of binding of Zur to znuAPro and ZntR to zntAPro were performed as described[29], by titration of Zn(II)-Zur into 1 μM znuAPro and Co(II)-ZntR into 2.5 μM zntAPro. DNA-binding affinities were determined using 10 nM dsDNA probe. DNA affinities and coupling free energies ($\Delta G_C$) were determined with Dynafit[29, 40, 47]. Models for RcnR and FrmR[E64H] have been described elsewhere[40]. Data for Zur were fit to a model describing sequential binding of two non-dissociable dimers to two sites on znuAPro. Data for ZntR were fit to a model describing binding of one non-dissociable dimer to zntAPro. The anisotropy change associated with a dimer binding to DNA was determined to be 0.025 by using Dynafit to simultaneously fit the data from apo-ZntR titrations ($n = 7$, Fig. 5f) and this value was then fixed to individually fit the data sets for apo-, Zn(II)- and Co(II)-ZntR and determine DNA affinities. Mean and standard deviation values were determined from at least triplicate analyses ('n' specified in figure legends).

**Quantitation of protein abundance.** E. coli strains BW25113ΔzntR/Δzur and BW25113ΔnikR/ΔrcnR, and Salmonella strain SL1344 were cultured to logarithmic phase in supplemented M9 minimal medium. Purified stocks of Zur, ZntR and RcnR were quantified by amino acid analysis (UC Davis). For ZntR and Zur quantitation, soluble cell lysates were prepared in 40 mM NaCl, 160 mM KCl, 10 mM EDTA, 10 mM DTT, 10 mM HEPES, pH 7.8, with addition of protease inhibitor mixture (Sigma), and post-sonication, the soluble cell lysate was syringe filtered (0.45-μm pore size), snap frozen in liquid N₂, stored at −80 °C, and thawed on ice before use[29]. Standard curve samples were prepared by dilution of purified protein stocks into cell lysates from BW25113ΔzntR/Δzur[29]. For RcnR quantitation, interference was observed using cell lysates directly, and an alternative approach was used. Following growth, harvested cells were stored at −20 °C before being resuspended in 300 mM NaCl, 10 mM EDTA, 1 mM TCEP, 10 mM HEPES pH 7.0, 1 mM PMSF. Standard curves were prepared by dilution of purified RcnR into soluble cell lysates from BW25113ΔnikR/ΔrcnR. Cell lysates (standard curve and experimental samples) were enriched for RcnR using a 1 ml HiTrap Heparin column (GE Healthcare) equilibrated in 300 mM NaCl, 10 mM EDTA, 1 mM TCEP, 10 mM HEPES pH 7.0. Bound protein was washed in binding buffer, and eluted in a single step in the same buffer but with addition of 1 M NaCl. Aliquots were stored at −80 °C. Heavy labelled peptides ([¹³C₆,¹⁵N₄]arginine residues; Thermo Fisher) were used as working internal standards (IS). Samples were prepared and analysed as described[29]. Briefly, samples were methanol precipitated, suspended in NH₄HCO₃ in 10% (v/v) methanol before addition of IS, and tryptic digestion. For the analysis of RcnR, methanol precipitation following enrichment was not required. Following centrifugation and solvent removal, samples were separated by gradient elution using a Zorbax Eclipse Plus C18 column and analysed by liquid chromatography tandem mass spectrometry operating in MRM mode. Mobile phase A and B consisted of 0.1% (v/v) formic acid in water and 0.1% (v/v) formic acid in acetonitrile, respectively. Aliquots were applied to a 6500 triple quadrupole mass spectrometer (AB Sciex) operating in positive ionisation mode. Acquisition methods used the following parameters: 5500 V ion spray voltage; 25 p. s.i. curtain gas; 60 p.s.i. source gas; 550 °C interface heating temperature; 40 V declustering potential; 26 V collision energy; and 27 V collision cell exit potential. Scheduled MRM was carried out with a 90 s multiple reaction monitoring detection window and 1.00 s target scan time. A quadratic 1/x² weighted regression model was used to perform standard calibration. Multiple peptides and transitions were initially assessed for each protein. The final transitions monitored were: 765.4/746.2 for Zur peptide ETEPQAKPPTIYR (770.4/756.2 for IS), 550.8/601.3 for ZntR peptide LADVTPDTIR (555.8/611.3 for IS), and 409.2/590.3 for RcnR peptide GAVNGLMR (414.2/600.3 for IS). MRM data across all experiments are presented in Supplementary Tables 1–3. Mean and standard deviations were determined from triplicate analyses (independently grown cultures).

**Mathematical modelling.** Fractional occupancy of DNA targets with sensor, as a function of buffered [metal] ([M]ᵦ), was modelled using Dynafit and the template script in Supplementary Software, using determined metal affinities ($1/K_1$), DNA

affinities ($1/K_3$ and $1/K_4$) plus cellular abundance of each sensor (P) and DNA target (D) (Table 1)[47]. Where the standard deviations for the DNA affinities of Co (II)- and Zn(II)-bound proteins overlapped, average values generated by combining the data for both metals were used for $1/K_4$. These were: $3.6 \times 10^{-8}$ M for Co (II)-Zur and Zn(II)$_2$-Zur, $1.4 \times 10^{-5}$ M for Co(II)-RcnR and Zn(II)-RcnR, and $4.7 \times 10^{-7}$ M for Co(II)-ZntR and Zn(II)-ZntR. To determine the amount of (P•M) •D for ZntR, the response for 'PD' was removed from the Dynafit script. A cell volume of 1 fl was used to calculate $[P]_{total}$ and $[D]_{total}$ from the number of protein assemblies per cell (i.e. dimers or tetramers) (Table 1) and target DNA binding sites per cell (assumed to be 1 copy per cell for RcnR and ZntR, 4 copies per cell for Zur due to additional gene targets[2], and 15 copies per cell for FrmR and FrmR$^{E64H}$ due to the presence of a low copy number reporter plasmid)[29]. Supplementary Data 1 and Supplementary Software provide Dynafit script and Excel spreadsheet to enable the above calculations. DNA occupancy by each sensor was normalised from zero to one using the minimum and maximum DNA occupancy values. FrmR and FrmR$^{E64H}$ were normalised to the same scale. The equilibrium concentration of free metal (i.e. [buffered metal]) was calculated using the equilibrium $M + B \rightleftharpoons MB$, where M = metal and B = buffer component. This was solved, via a quadratic equation, using initial concentrations of metal and buffer of 0.01 and 0.1 M, respectively, and each equilibrium constant ($K_5$) value listed in Supplementary Data 1. The following are step-by-step instructions to calculate fractional DNA occupancy with sensor as a function of buffered [metal] using Supplementary Data 1 and Supplementary Software: (1) Input the number of protein assemblies (functional units i.e. dimer/tetramer), DNA binding sites and cell volume into the blue boxes within Supplementary Data 1. (2) Input association constants $K1$, $K3$ and $K4$ into the green boxes within Supplementary Data 1. (3) Open Dynafit and load script file Supplementary Software. Edit the following parameters 'M', 'D', 'B', 'Keq1-4' and responses 'PD' and 'PMD', using the values in the orange boxes from Supplementary Data 1. (4) Adjust 'Keq5' to 1e-1. (5) Edit [set:P] with the contents of 'G6' and 'H6' from Supplementary Data 1. (6) Edit location of Output file as appropriate. Currently C:/ (7) Run script. (8) Open output file 'data-model-t001-s001.txt' and obtain value for 'y(data)'. Input into cell 'N3' in Supplementary Data 1. This gives the fractional occupancy of DNA at a buffered metal concentration of $5.12 \times 10^{-3}$ M. (9) Repeat Dynafit simulation a further fourteen times. Each time, adjust 'Keq5' using values given in Supplementary Data 1 column 'L', which will achieve corresponding buffered metal concentrations in column 'M'. Complete column 'N' as described in step 8. (10) Plot fractional DNA occupancy (column 'N') as a function of buffered metal concentration (column 'M') and adjust the $x$ axis to logarithmic display. (11) Fractional DNA occupancy can then be normalised to a scale from zero to one if required.

**Code availability**. Supplementary Data 1 and Supplementary Software include a Dynafit script and template Excel spreadsheet to enable calculation of promoter occupancy by a metal sensor as a function of buffered metal concentration, with detailed step-by-step instructions described in the Methods section.

**Data availability**. The data supporting the findings of this study are available within the article and its Supplementary Information files, or from the corresponding author upon request.

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

## Acknowledgements

This work was supported by Biotechnology and Biological Research Council awards BB/J017787/1 and BB/L009226/1 to N.J.R. in conjunction with a financial contribution by Procter and Gamble (in association with BB/J017787/1 Industrial Partnership Award). We thank DBS Genomics for sequencing of plasmid constructs, D. Weinkove for *E. coli* deletion strains, P.T. Chivers for P1 lysate from BW25113Δ*rcnR::kan* and E. Pohl for help with structural homology modelling (all Durham University).

## Author contributions

D.O. conducted the in vivo expression experiments, bioinformatic analysis, in vitro characterisation of Zur and FrmR[E64H]. A.W.F. determined the Co(II) affinity of ZntR and DNA affinity of Zn(II)-RcnR. A.W.F., with help from J.W.S. and D.O. developed the Dynafit script to determine fractional DNA occupancy with metal sensor. J.C. and T.G.H. performed the MRM tandem mass spectrometry. K.S. determined the Zn(II) affinity of RcnR and DNA affinities of ZntR. N.J.R. and E.L.-L. were responsible for the conception of the programme. N.J.R. and D.O. drafted the manuscript and with A.W.F. interpreted the significance of the data. N.J.R. had overall responsibility for the design and co-ordination of the programme. All authors reviewed the results and edited and approved the final version of the manuscript.

## Additional information

**Competing interests:** This work was supported by Biotechnology and Biological Research Council awards BB/J017787/1 and BB/L009226/1 to N.J.R. in conjunction with a financial contribution by Procter and Gamble (in association with BB/J017787/1 Industrial Partnership Award). The remaining authors declare no competing financial interests.

