## [Peer Review File · Nature Communications]

Reviewers' comments:

Reviewer #1 (Remarks to the Author):

The manuscript by Osman et al. entitled "Fine control of metal concentrations is necessary for cells to discern zinc from cobalt" describes their findings on the sensing and responsiveness of various *Salmonella* transcriptional regulators, Zur, ZntR, RcnR, FrmR and a FrmR mutant (FrmR_E64H) to cobalt and zinc stress. The authors determined the affinities of the recombinant proteins for the metal ligands and their DNA targets as well as their couple free energies and relative cellular abundance. They then use these data in a mathematical model of metal sensing and correlate the outcomes to the transcriptional responsiveness of their targets under stress. Overall, their study describes how metal stress could cause mismetallation of transcription factors and hence the dysregulation of metal homeostatic mechanisms.

This is a very need study, with a lot of high-quality and well-presented data, in particular the recombinant protein work. The questions posed are highly relevant to the field and the findings have significant potential for a broader impact across cell and metal biology. One of the key distinctions of their study to most previous analyses of metal toxicity is the in-depth examination of metal shock treatment, which will influence the design future studies in metal toxicity.

Major comments:

1. Including novel data in the introduction is confusing. To highlight the significance of their findings, the authors should first focus on describing already published works on metal toxicity, intracellular metal buffering, mismetallation and dysregulation of (bacterial) metal sensors. There are many similar studies and although they are cited in very broad comments about these topics, the background has to be written in greater detail. This would then excite the reader about the novelty of their current findings.

2. I do not understand why the authors present non-quantitative RT-PCR data. These data sets play a key role in the validation of their mathematical model and their RT-PCR data is extremely subjective. Quantitative RT-PCR assays are technically straightforward and allow for subsequent statistical analyses. I believe this is essential for this manuscript as a lot of emphasis has been put on the "trends" (as the authors have denoted their RT-PCR results on page 11 line 22), these subjective trends have than been indicted by arrowheads in Figures 2b and 9b. Furthermore, the analyses for fig 1d should be performed using the same methods as those for fig 1a-c.

3. This paper does not make any effort in examining potential downstream effects of mismetallation/dysregulation under metal stress. Is cellular metal ion homeostasis affected or longer-term tolerance to metal stress?

Minor comments

1. The difference between examining cells by continuous or shock treatment was not highlighted enough and the biological relevance of metal toxicity in any form has not been introduced appropriately.

2. Page 1: Include info on the salmonella-specific zinc and cobalt homeostasis mechanisms.

3. Page 1 Line 30: Remove "The latter"

4. Page 2 line 5: Is cobalt written in full because of its multiple oxidation states?

5. Page 2 line 25: Remove this line, it doesn't make sense here.

6. Page 2 line 34: The headings are inconsistent, some are conclusions and some specify the nature of the analysis.

7. Page 2 line 37: The data on how concentrations for cellular assays were determined is required (growth curves?). Can the authors also elaborate on metal buffering and why salmonella can tolerate far greater levels of zinc over cobalt?
8. Page 3 line 4: This figure show repression of *znuA* upon zinc supplementation, please reword.
9. Page 3 lines 8-15 (fig 1d): Instead of making the data comparable with previous studies, the data should at least be comparable with other analyses in the current study (fig 1a-c).
10. Page 3 lines 17-35: All of this requires quantitation.
11. Page 5 line 19: The oligonucleotides are not listed in Table S4. Include citation in table if sequences from a previous study were used.
12. Page 5 line 23: Figs. 1d should be 1b.
13. Page 5 line 27: Please elaborate on the role of Zn and Co in the second binding event considering the results seen in Figs 4 f,g,h.
14. Page 9 lines 25-37: This section seems more appropriate in the introduction.

Reviewer #2 (Remarks to the Author):

Overview.

This is an outstanding manuscript both in terms of the experimental design and its and the description of the outcomes. Although complex, the manuscript is easily readable, clearly conveying the rationale of an experiment, its outcome and its significance in relation to the objectives of this research: how can you explain the metal specificity of a trans-factor that in vitro is relatively promiscuous as to the divalent metal it binds. Given the plethora of such metal sensors and their critical roles in bacterial homeostasis and, commonly, bacterial pathogenesis, this selectively is a significant unknown giving this research a wide audience. In this regard, the subject matter of this work is widely suitable for publication in a journal like Nat. Comm. Last, the rigor of the experimental design and statistical analysis of the data is excellent and reflects the quality expected in a Nature publication.

Specific concern that needs to be addressed.

The authors make a critical contribution by explicitly taking into account the (typically) positive cooperativity in metal binding in the context of sensor binding to its operator site. Assuming that a mis-metalated sensor doesn't bind to DNA, this by itself promotes selectivity. But the authors add another wrinkle, also of significance, and that is a "buffer" component that modulates the metal available to the sensor; their simulations work because of this "buffer" and while I am not at all worried about inclusion of this component in their "cell," at the same time, given the lack any discussion of what this "buffer" might be, it comes close to representing a "fudge factor." In short, their model indicates that what we should be examining is this "buffer" as it comes between metal uptake and metal binding to a sensor molecule. Is this buffer uniformly distributed in the cell? Is it solely cytoplasmic or solely vesicular? Or both? What kind of vesicles? Is it linked in any way to uptake transporters? How is it distributed with respect to the sensors? Is it functioning in the nucleus? Last and perhaps most important, is there only one divalent metal ion buffer or are there buffers specific to a metal ion? Isn't this pushing the question about selectively up-stream from the sensor question?

Perhaps I'm missing something, perhaps I'm asking more from this manuscript than intended by the authors. On the other hand, if I have these questions, other readers will, too. They need to be

explicitly considered and addressed.

Reviewer #3 (Remarks to the Author):

In their manuscript entitled, 'Fine control of metal concentrations is necessary for cells to discern zinc from cobalt,' Robinson and co-workers try to address a conundrum in the field of metal sensing proteins. There are a suite of metal sensing proteins in all bacteria (also called metalloregulatory proteins), that function to regulate specific metal homeostasis in response to metal binding – for example, in some cases the protein is inactive in the absence of metal ion coordination and then is activated to regulate certain genes in response to metal ion coordination. The modes of activation differ (co-repression, conformational change, etc); however the basic principle of +/- metal binding leading to change in regulation applies to all. The conundrum concerns the question how does a specific metalloregulatory protein respond to a specific metal ion. If several metalloregulatory proteins that are known to regulate different metal ions in cells, e.g. zinc, iron and nickel, are over-expressed and purified in vitro, they will invariably all bind zinc with a higher affinity in vitro because binding affinity in vitro follows the Irving-Williams series. How then do the proteins that are supposed to regulate iron, nickel, cobalt etc in cells manage to regulate these metals, if zinc will always bind preferentially. Robinson and co-workers have pioneered the idea that in cells the metal selectivity of metalloregulators is not just a product of simple thermodynamics, and have taught us that issues such as local metal availability, buffering capacity, compartmentalization, etc. are important for metal selectivity.

In this manuscript, this question is further investigated by examining several sensors found in *Salmonella Typhimurium* - Zur, ZntR (both Zn sensors), RcnR (a Co sensor) and FrmR and – a formaldehyde sensor that can sense metals in vitro and be modified to sense Zn and Co in vivo (E64H). Robinson and co-workers present a very comprehensive set of data, collected both in cells and in vitro (metal binding and DNA binding) that allows them to propose and test rigorous thermodynamic model that should allow one to predict how a given metalloregulatory protein will respond to metal ions in cells. The model, given in Figure 6 (and fit in the program Dynafit – with explicit details presented in supplementary), takes into account all of the species that can be present in the cell to determine which metal a protein will respond to. A key parameter that is included is the concentration of buffered metal in a cell. This is defined hypothetically [B], such that the [M] (metal concentration) is >1000 fold higher than the [P] (protein concentration) and 10 times lower than the [B] (buffered metal concentration). Under these conditions, there is a surplus of metal at a given buffer concentration. Using this approach, along with either data collected here or with previously collected data, that includes metal affinity, DNA affinity of apo-protein, DNA affinity of metallated protein, and cellular abundance of the protein (determined by MRM tandem mass spectrometry), the group is able to demonstrate how a metal sensor is selective for a specific metal despite that metal being thermodynamically unfavored in simple in vitro conditions. I found the data that showed that RcnR was the only protein in the suite studied that would respond under low Co concentrations particularly compelling, as it really helped explain metal selectivity in cells. This model appears to be very useful as it allows one to determine how the local metal concentrations affect the activation of multiple metal sensor proteins. Overall, this is an important paper that merits publication in Nature Communications. I imagine that many researchers in the field will use this thermodynamic profile to understand metal selectivity of their own proteins.

A few questions that may be addressed to further clarify the paper:

1. What is known about how local metal concentrations vary in a cell? The premise of this work is

that it the local concentration of buffered metal determined which metal the metal sensor responds to, and a hypothetical metal buffering concentration is used. How would one think about measuring the local metal concentration in a cell?

2. Several studies in which metal – protein – DNA binding were measured by fluorescence anisotropy. In all cases the metal was loaded into the protein and then affinity to DNA was measured, or alternately the apo protein was studied. An interesting experiment may be to titrate an alternate metal with the metal-protein-DNA complex, e.g. what happens if Co is titrated with the Zn-Zur DNA complex and/or Zn into the Co-Zur DNA complex. Have the authors looked at this and/or could they comment on how this interaction may also affect their model?

The figures and schemes were well presented; however, in several cases the figures were very small and hard to discern. For example, in figure 1, all of the protein structures are really tiny and its hard to make out the metal sites. I would recommend enlarging some of the font/figure elements.

Point-by-point responses to the referees' comments (separate from cover letter):

Reviewer #1 (Remarks to the Author):

The manuscript by Osman et al. entitled "Fine control of metal concentrations is necessary for cells to discern zinc from cobalt" describes their findings on the sensing and responsiveness of various Salmonella transcriptional regulators, Zur, ZntR, RcnR, FrmR and a FrmR mutant (FrmR_E64H) to cobalt and zinc stress. The authors determined the affinities of the recombinant proteins for the metal ligands and their DNA targets as well as their couple free energies and relative cellular abundance. They then use these data in a mathematical model of metal sensing and correlate the outcomes to the transcriptional responsiveness of their targets under stress. Overall, their study describes how metal stress could cause mismetallation of transcription factors and hence the dysregulation of metal homeostatic mechanisms.

This is a very need study, with a lot of high-quality and well-presented data, in particular the recombinant protein work. The questions posed are highly relevant to the field and the findings have significant potential for a broader impact across cell and metal biology. One of the key distinctions of their study to most previous analyses of metal toxicity is the in-depth examination of metal shock treatment, which will influence the design future studies in metal toxicity.

Major comments:

1. Including novel data in the introduction is confusing.

The gene expression data have been removed from introductory Fig. 1 (Page 24).

To highlight the significance of their findings, the authors should first focus on describing already published works on metal toxicity, intracellular metal buffering, mismetallation and dysregulation of (bacterial) metal sensors. There are many similar studies and although they are cited in very broad comments about these topics, the background has to be written in greater detail. This would then excite the reader about the novelty of their current findings.
The introduction has been edited with a new second and fourth paragraph covering published work on metal toxicity, mismetallation, metal-buffering, regulation and dysregulation of bacterial metal sensors (Page 2, lines 4 to 25; Page 3 lines 7 to 21). A new third paragraph describing intracellular metal buffering in detail has been added to the discussion to simultaneously address the second part of minor comment 7 plus suggestions of referees 2 and 3 (Page 11, line 8 to Page 12, line 3). In making these additions care has been taken to retain a focussed narrative suitable for non-specialist readers.

2. I do not understand why the authors present non-quantitative RT-PCR data. These data sets play a key role in the validation of their mathematical model and their RT-PCR data is extremely subjective. Quantitative RT-PCR assays are technically straightforward and allow for subsequent statistical analyses. I believe this is essential for this manuscript as a lot of emphasis has been put on the "trends" (as the authors have denoted their RT-PCR results on page 11 line 22), these subjective trends have than been indicted by arrowheads in Figures 2b and 9b.

We very much appreciate this suggestion. We had not previously used quantitative PCR and, as stated by the referee it was in fact technically straightforward to do. Quantitative PCR establishes the trends previously indicated by end-point PCR in the original Figs. 1, 2b and 9b, and now shown in the new Figs. 2, 3 and 9c (Pages 26, 27 and 33) plus associated Supplementary Figures.

Furthermore, the analyses for fig 1d should be performed using the same methods as those for fig 1a-c.

The data that were in Fig. 1d have been replaced (now Fig. 2d-f, Page 26) using primers corresponding to the *frm* operon for both end point PCR and for quantitative PCR. Expression data under the control of FrmR and FrmR^{E64H}, previously in Figs. 2 and 9, have also been replaced (now in Figs. 3 and 9; Pages 27 and 33).

3. *This paper does not make any effort in examining potential downstream effects of mismetallation/dysregulation under metal stress. Is cellular metal ion homeostasis affected or longer-term tolerance to metal stress?*

Growth data have been added as Supplementary Figs. 2 and 11. Supplementary Fig. 11 shows that prolonged exposure to the higher metal concentrations used in the shock treatments do indeed become inhibitory, indicative of downstream effects on, for example, protein mis-metalation. Notes to this effect have been added to the results (Page 4, lines 32 to 34; Page 5, lines 12 to 13; Page 9, lines 23 to 24).

Minor comments:

1. *The difference between examining cells by continuous or shock treatment was not highlighted enough and the biological relevance of metal toxicity in any form has not been introduced appropriately.*

The new second paragraph added to the introduction highlights differences between continuous and shock treatments, plus introduces metal toxicity and mis-metalation (Page 2, lines 4 to 26).

2. *Page 1: Include info on the salmonella-specific zinc and cobalt homeostasis mechanisms.*

The roles of the proteins encoded by the *Salmonella*-specific zinc and cobalt homeostatic mechanisms have been added to the introduction (Page 2, lines 30 to 34), along with descriptions of the respective regulators (Page 1, lines 36 to 37; Page 2, lines 27 to 29), as also illustrated in the revised Fig. 1 (Page 24).

3. *Page 1 Line 30: Remove "The latter"*

This has been removed.

4. *Page 2 line 5: Is cobalt written in full because of its multiple oxidation states?*

Yes: Where the oxidation state of a metal is known it is specified and where it is unknown the name of the metal is written in full.

5. *Page 2 line 25: Remove this line, it doesn't make sense here.*

This line has been removed. To help readers to appreciate an advance described in this manuscript, and to encourage use of the methodology by others (cross refer to first response to reviewer #3), the following has been added to Page 3, line 26: "*The computational methods are set out in a format to assist their use by others*".

6. *Page 2 line 34: The headings are inconsistent, some are conclusions and some specify the nature of the analysis.*

The headings have been standardised as conclusions/observations.

7. *Page 2 line 37: The data on how concentrations for cellular assays were determined is required (growth curves?).*

The metal concentrations used for prolonged exposures were selected to be the maximum giving negligible ($\leq 10\%$) inhibition of growth at mid-logarithmic phase (the stage at which the assays were performed) and these data have now been added as Supplementary Fig. 2 and 11, and referred to in the results text (Page 4, line 5; Page 4, line 32 to 34; Page 5, lines 12 to 13; Page 9, lines 23 to 24), along with growth and survival data for the shock treatments (Supplementary Fig. 11).

Can the authors also elaborate on metal buffering and why salmonella can tolerate far greater levels of zinc over cobalt?

An additional third paragraph of the discussion (Page 11, line 8 to page 12, line 3) elaborates on metal buffering, notes that *Salmonella* tolerates far greater levels of Zn(II) over cobalt (now shown in Supplementary Fig. 11) and comments on the fact that *Salmonella*, unlike *E. coli*, uses cobalt to make cobalamin (vitamin B₁₂), but only under anaerobic conditions with implications for the toxicity of cobalt in aerobic cultures.

8. Page 3 line 4: This figure show repression of znuA upon zinc supplementation, please reword.

This has been reworded as suggested (now on Page 4). The expression of the targets of other regulators has similarly been reworded. These results are also supported by the new quantitative PCR data which has been added to the revised Fig. 2 (Page 26).

9. Page 3 lines 8-15 (fig 1d): Instead of making the data comparable with previous studies, the data should at least be comparable with other analyses in the current study (fig 1a-c).

This has been changed as suggested: See response to the second part of major comment 2 above.

10. Page 3 lines 17-35: All of this requires quantitation.

This has been done by quantitative PCR as suggested, and the text modified.

11. Page 5 line 19: The oligonucleotides are not listed in Table S4. Include citation in table if sequences from a previous study were used.

The oligonucleotides used in previous studies have now been added to Supplementary Table 4 along with the relevant citations.

12. Page 5 line 23: Figs. 1d should be 1b.

This has been corrected.

13. Page 5 line 27: Please elaborate on the role of Zn and Co in the second binding event considering the results seen in Figs 4 f,g,h.

The following has been added to the text (Page 7, lines 8 to 10): “Both Co(II) and Zn(II) encouraged formation of ternary complexes, at least on this (34 bp) DNA fragment.” With longer DNA (44 bp) we do not see this effect of Co(II) and Zn(II), hence the caveat in the text.

14. Page 9 lines 25-37: This section seems more appropriate in the introduction.

This text has been moved to the introduction (Page 2, lines 4 to 25).

Reviewer #2 (Remarks to the Author):

Overview.

This is an outstanding manuscript both in terms of the experimental design and its and the description of the outcomes. Although complex, the manuscript is easily readable, clearly conveying the rationale of an experiment, its outcome and its significance in relation to the objectives of this research: how can you explain the metal specificity of a trans-factor that in vitro is relatively promiscuous as to the divalent metal it binds. Given the plethora of such metal sensors and their critical roles in bacterial homeostasis and, commonly, bacterial pathogenesis, this selectively is a significant unknown giving this research a wide audience. In this regard, the subject matter of this work is widely suitable for publication in a journal like Nat. Comm. Last, the rigor of the experimental design and statistical analysis of the data is excellent and reflects the quality expected in a Nature publication.

Specific concern that needs to be addressed:

The authors make a critical contribution by explicitly taking into account the (typically) positive cooperativity in metal binding in the context of sensor binding to its operator site. Assuming that a mis-metalated sensor doesn't bind to DNA, this by itself promotes selectivity. But the authors add another wrinkle, also of significance, and that is a "buffer" component that modulates the metal available to the sensor; their simulations work because of this "buffer" and while I am not at all worried about inclusion of this component in their "cell," at the same time, given the lack any discussion of what this "buffer" might be, it comes close to representing a "fudge factor." In short, their model indicates that what we should be examining is this "buffer" as it comes between metal uptake and metal binding to a sensor molecule. Is this buffer uniformly distributed in the cell? Is it solely cytoplasmic or solely vesicular? Or both? What kind of vesicles? Is it linked in any way to uptake transporters? How is it distributed with respect to the sensors? Is it functioning in the nucleus? Last and perhaps most important, is there only one divalent metal ion buffer or are there buffers specific to a metal ion? Isn't this pushing the question about selectively up-stream from the sensor question? Perhaps I'm missing something, perhaps I'm asking more from this manuscript than intended by the authors. On the other hand, if I have these questions, other readers will, too. They need to be explicitly considered and addressed.

These questions have been directly addressed in a new third paragraph which has been added to the discussion and helps to make a more complete narrative (Page 11, line 8 to Page 12, line 3).

Reviewer #3 (Remarks to the Author):

In their manuscript entitled, 'Fine control of metal concentrations is necessary for cells to discern zinc from cobalt,' Robinson and co-workers try to address a conundrum in the field of metal sensing proteins. There are a suite of metal sensing proteins in all bacteria (also called metalloregulatory proteins), that function to regulate specific metal homeostasis in response to metal binding – for example, in some cases the protein is inactive in the absence of metal ion coordination and then is activated to regulate certain genes in response to metal ion coordination. The modes of activation differ (co-repression, conformational change, etc); however the basic principle of +/- metal binding leading to change in regulation applies to all. The conundrum concerns the question how does a specific metalloregulatory protein respond to a specific metal ion. If several metalloregulatory proteins that are known to regulate different metal ions in cells, e.g. zinc, iron and nickel, are over-expressed and purified in vitro, they will invariably all bind zinc with a higher affinity in vitro because binding affinity in vitro follows the Irving-Williams series. How then do the proteins that are supposed to regulate iron, nickel, cobalt etc in cells manage to regulate these metals, if zinc will always bind preferentially. Robinson and co-workers have pioneered the idea that in cells the metal selectivity of metalloregulators is not just a product of simple thermodynamics, and have taught us that issues such as local metal availability, buffering capacity, compartmentalization, etc. are important for metal selectivity.

In this manuscript, this question is further investigated by examining several sensors found in Salmonella Typhimurium - Zur, ZntR (both Zn sensors), RcnR (a Co sensor) and FrmR and – a formaldehyde sensor that can sense metals in vitro and be modified to sense Zn and Co in vivo (E64H). Robinson and co-workers present a very comprehensive set of data, collected both in cells and in vitro (metal binding and DNA binding) that allows them to propose and test rigorous thermodynamic model that should allow one to predict how a given metalloregulatory protein will respond to metal ions in cells. The model, given in Figure 6 (and fit in the program Dynafit – with explicit details presented in supplementary), takes into account all of the species that can be present in the cell to determine which metal a protein will respond to. A key parameter that is included is the concentration of buffered metal in a cell. This is defined hypothetically [B], such that the [M] (metal concentration) is

>1000 fold higher than the $[P]$ (protein concentration) and 10 times lower than the $[B]$ (buffered metal concentration). Under these conditions, there is a surplus of metal at a given buffer concentration. Using this approach, along with either data collected here or with previously collected data, that includes metal affinity, DNA affinity of apo-protein, DNA affinity of metallated protein, and cellular abundance of the protein (determined by MRM tandem mass spectrometry), the group is able to demonstrate how a metal sensor is selective for a specific metal despite that metal being thermodynamically unfavored in simple *in vitro* conditions. I found the data that showed that RcnR was the only protein in the suite studied that would respond under low Co concentrations particularly compelling, as it really helped explain metal selectivity in cells. This model appears to be very useful as it allows one to determine how the local metal concentrations affect the activation of multiple metal sensor proteins. Overall, this is an important paper that merits publication in *Nature Communications*. I imagine that many researchers in the field will use this thermodynamic profile to understand metal selectivity of their own proteins.

We think that this is an important point, and so to encourage the use of this approach by other researchers, and as noted in response to reviewer #1, the following has been added to Page 3, line 26: “The computational methods are set out in a format to assist their use by others”.

A few questions that may be addressed to further clarify the paper:

1. *What is known about how local metal concentrations vary in a cell? The premise of this work is that it the local concentration of buffered metal determined which metal the metal sensor responds to, and a hypothetical metal buffering concentration is used. How would one think about measuring the local metal concentration in a cell?*

These two questions have been addressed in the new third paragraph of the discussion (Page 11, line 8 to Page 12, line 3).

2. *Several studies in which metal – protein – DNA binding were measured by fluorescence anisotropy. In all cases the metal was loaded into the protein and then affinity to DNA was measured, or alternately the apo protein was studied. An interesting experiment may be to titrate an alternate metal with the metal-protein-DNA complex, e.g. what happens if Co is titrated with the Zn-Zur DNA complex and/or Zn into the Co-Zur DNA complex. Have the authors looked at this and/or could they comment on how this interaction may also affect their model?*

This is an interesting suggestion best suited to a combination of metals in which one of the pair is not allosterically effective: Zn(II) and Co(II) both trigger allostery in these sensors. A tight binding metal which is not allosterically effective has the potential to block sensing of a weaker binding cognate metal, indeed this could be a component of nutritional immunity. Evidence that Fe(II) is antagonistic to Mn(II)-sensing by MntR in *Bacillus subtilis* has been added on Page 2, lines 8 to 9.

The figures and schemes were well presented; however, in several cases the figures were very small and hard to discern. For example, in figure 1, all of the protein structures are really tiny and its hard to make out the metal sites. I would recommend enlarging some of the font/figure elements.

The protein structures in Fig. 1 (Page 24) have been enlarged and (as noted earlier) separated from the data which is now in the new Fig. 2 (Page 26), and the metal sites have been further enlarged specifically. We have also removed some gratuitous lines in later figures to make them simpler, and have standardised the font size. To retain ten graphic items in the main text, the MRM chromatograms (former Fig. 5) have been moved to a new Supplementary Fig. 14.

During editing of the manuscript we were able to improve the modelled fits for Zur binding to DNA (Fig. 5b,c and Fig. 8d) generating more accurate K_{DNA} values in Table 1

and these were used to model promoter occupancy as a function of metal concentrations (Fig. 7 and Fig. 9), which will be (marginally) more accurate.

We thank all of the reviewers for encouraging us to make alterations, and additions which improve the manuscript.

REVIEWERS' COMMENTS:

Reviewer #1 (Remarks to the Author):

The authors have done an excellent job on the revised version of their manuscript and have addressed all my queries. Hence, I highly recommend publication of this manuscript in Nature Communications and congratulate the authors on their outstanding work.

Reviewer #3 (Remarks to the Author):

The authors have satisfactorily addressed my comments in the revised version. I recommend publication of this work.

REVIEWERS' COMMENTS:

Reviewer #1 (Remarks to the Author):

The authors have done an excellent job on the revised version of their manuscript and have addressed all my queries. Hence, I highly recommend publication of this manuscript in Nature Communications and congratulate the authors on their outstanding work.

Reviewer #3 (Remarks to the Author):

The authors have satisfactorily addressed my comments in the revised version. I recommend publication of this work.

We thank both the reviewers for their positive comments and for earlier constructive suggestions which encouraged us to improve the manuscript.